



# Temperature and water vapour measurements in the framework of NDACC

Benedetto De Rosa, Paolo Di Girolamo, Donato Summa

Scuola di Ingegneria, Università degli Studi della Basilicata, Potenza, 85100, Italy

*Correspondence to*: Paolo Di Girolamo (paolo.digirolamo@unibas.it)

**Abstract.** The Raman Lidar system BASIL entered the International Network for the Detection of Atmospheric Composition Change (NDACC) in 2012. Since then measurements were carried out routinely on a weekly basis. This manuscript reports specific measurement results from this effort, with a dedicated focus on temperature and water vapour profile measurements. The main objective of this research effort is to provide a characterization of the system performance. Measurements

illustrated in this manuscript demonstrate the ability of BASIL to perform measurements of the temperature profile up to 50 km and of the water vapour mixing ratio profile up to 15 km, when considering an integration time of 2 h and a vertical resolution of 150 m, with measurement bias not exceeding 0.1 K and 0.1 g kg$^{-1}$, respectively. Relative humidity profiling capability up to the tropopause is also demonstrated by combining simultaneous temperature and water vapour profile measurements.

Raman lidar measurements are compared with measurements from additional instruments, such as radiosondings and satellite sensors (IASI and AIRS), and with model re-analyses data (ECMWF and ECMWF-ERA). Comparisons in this paper cover the altitude region up to 15 km for water vapour mixing ratio and up to 50 km for the temperature. We focused our attention on four selected case studies collected during the first 2 years of operation of the system (November 2013-October 2015). Comparisons between BASIL and the different sensor/model data in terms of water vapour mixing ratio

indicate a mean absolute/relative bias of -0.024 g kg$^{-1}$ (or -3.9 %), 0.346 g kg$^{-1}$ (or 37.5 %), 0.342 g kg$^{-1}$ (or 36.8 %), -0.297 g kg$^{-1}$ (or -25 %), -0.381 g kg$^{-1}$ (or -31 %), when compared with radisondings, IASI, AIRS, ECMWF, ECMWF-ERA, respectively. For what concerns the comparisons in terms of temperature measurements, these reveal a mean absolute bias between BASIL and the radiosondings, IASI, AIRS, ECMWF, ECMWF-ERA of -0.04, 0.48, 1.99, 0.14, 0.62 K, respectively. Based on the available dataset and benefiting from the circumstance that the Raman lidar BASIL could be compared with all

other sensor/model data, it was possible to estimate the absolute bias of all sensors/datasets, this being 0.004 g kg$^{-1}$/0.30 K, 0.021 g kg$^{-1}$/-0.34 K, -0.35 g kg$^{-1}$/0.18 K, -0.346 g kg$^{-1}$/-1.63 K, 0.293 g kg$^{-1}$/-0.16 K and 0.377 g kg$^{-1}$/0.32 K for the water vapour mixing ratio/temperature profile measurements carried out by BASIL, the radiosondings, IASI, AIRS, ECMWF, ECMWF-ERA, respectively.



## 1 Introduction

Water vapour is the most important atmospheric greenhouse gas and its increasing tropospheric concentration, though indirectly, is driven primarily by human activities. Increasing concentrations of $CO_2$ and $CH_4$, primarily associated with fossil fuel combustion, lead to warmer tropospheric temperatures, which are responsible for increased atmospheric humidity
contents and ultimately lead to a warmer climate (IPCC, 2007). Water vapour in the upper troposphere/lower stratosphere (UTLS) region has a crucial role in the Earth radiative budget, and consequently in the climate system. Its presence at these altitudes being primarily associated with two main sources: transport from the troposphere, taking place mainly in the tropics, and the in-situ oxidation of methane. Temperature and water vapour concentration changes in the UTLS result in radiative forcing alterations (among others, Riese *et al.*, 2012). Observations demonstrated that stratospheric water vapour
concentration increases with increasing tropospheric temperature, implying the existence of a stratospheric water vapour feedback (Dessler *et al.*, 2013). The strength of this feedback has been estimated to be ~0.3 W m$^{-2}$ K$^{-1}$ (Dessler *et al.*, 2013). Stratospheric water vapour has also an important role in stratospheric clouds formation, which are a key element in stratospheric ozone depletion mechanisms (di Sarra *et al.*, 1992, Di Girolamo *et al.*, 1994). Furthermore, stratospheric water vapour has also a primary importance in the processes leading to the formation of hydrogen radicals, and consequently in
stratospheric chemistry and ozone depletion mechanisms (Lossow *et al.*, 2013).

Despite the well-recognized importance of having accurate tropospheric and stratospheric water vapour and temperature profile measurements, data sets of these variables and their long-term variability are limited, especially in the UTLS region. Quality water vapour measurements in the UTLS region are provided by radiosondes or balloon-borne frost-point hygrometers This latter is considered to be the most accurate water vapour sensor for the low humidity levels found in the
UTLS region (Vomel *et al.*, 2007). However, the global radiosonde network, including ~800 stations, is quite sparse and with limited coverage in oceanic areas. Additionally, radiosondes are quite expensive and their operational launch schedule (typically two or four times per day) use, is not sufficiently intense to guarantee the temporal resolution required for the above mentioned scientific scopes. Water vapour measurements by satellite limb sounders, both in the infrared and microwave domain, have demonstrated to lack both time and horizontal resolution (Griessbach *et al.*, 2016, Hurst *et al.*,
2014). Similar considerations apply to temperature profiling, the main source of measurements covering the upper troposphere and the stratosphere being microwave and infrared satellite sounders (Thorne *et al.*, 2005).

All the above weather and climate-related issues call for highly accurate measurements of both the water vapour and temperature profiles throughout the troposphere and stratosphere, with a specific focus on the UTLS region. These motivations pushed the Network for the Detection of Atmospheric Composition Change (NDACC), formerly the
international Network for the Detection of Stratospheric Change (NCSC), to include in the early 2000s water vapour and temperature lidars among its ensemble of instruments. NDACC, originally focussing on the long-term monitoring of stratospheric changes and ozone, has progressively broadened its priorities to include the monitoring of other atmospheric species and assessing their impacts on the stratosphere and troposphere. Climate and atmospheric composition changes have



a significant impact of the atmospheric thermal structure and this makes atmospheric temperature measurements of paramount importance for NDACC.

The University of BASILicata Raman Lidar system (BASIL) entered NDACC in November 2012. The primary contribution of BASIL to NDACC is to provide accurate routine measurements of the vertical profiles of both water vapour mixing ratio

and temperature. Temperature profile measurements by BASIL cover the altitude interval from surface up to the stratopause (~ 50 km). Measurements over such a wide altitude interval are possible based on the combined use of the pure rotational Raman technique (Behrendt and Reichardt, 2000), which allows covering the lowest 20 km, and the integration technique (Hauchercorne *et al.*, 1992), covering the altitude region from 20 km to typically 50-55 km. The combined application of these two techniques is possible because of the presence of an overlap region (20-25 km) where both techniques properly

work. Recently, the aerosol backscattering coefficient at 354.7 nm has been added to the set of atmospheric variables measured by BASIL and made available to the international community through the NDACC repository.

In the present research work we illustrate and discuss temperature and water vapour profile measurements from BASIL with the purpose of assessing system performance in terms of measurement BIAS. Specific measurement examples are considered for this effort, which are compared with measurements from other instruments, such as radiosondings and satellite sensors

(IASI and AIRS), and with model re-analyses data (ECMWF and ECMWF-ERA).

The paper outline is as follows. Section 2 gives a brief description of the Raman lidar set-up and its operation schedule in the frame of NDACC. Section 3 describes the additional profiling sensors and model data involved in the present inter-comparison effort. Section 4 illustrates the different lidar techniques considered to measure atmospheric thermodynamic variables, while section 5 defines the statistical quantities used in the inter-comparison for the assessment of the

measurement performance. Section 6 illustrates the inter-comparison results and provides an assessment of the performance of the considered sensors and models. Finally, section 7 summarizes all reported results and illustrates some possible future developments of the present study.

**2 The Raman lidar BASIL and its operation in the frame of NDACC**

The Network for the Detection of Atmospheric Composition Change (NDACC) became operational in 1991.It includes more

than 70 globally distributed, ground-based remote-sensing research stations for the observation of the physical and chemical state of the upper troposphere and stratosphere and their changes and for assessing the impact of these changes on global climate. Trends in the chemical and physical state of the atmosphere can be detected based on the collection of long-term databases. NDACC includes approx.. 25 ground-based lidar systems distributed worldwide, which are routinely operated for the monitoring of atmospheric temperature, ozone, ozone, aerosols, water vapour and polar stratospheric clouds. To extend

its research, NDACC has also established formal collaboration agreements with other eight major research networks (De Mazière *et al.*, 2018), namely: the AErosol RObotic NETwork (AERONET), the Baseline Surface Radiation Network (BSRN), the Advanced Global Atmospheric Gases Experiment (AGAGE), the Global Climate Observing System (GCOS) Reference Upper-Air Network (GRUAN), the National Aeronautics and Space Administration (NASA) Micro Pulse Lidar





Network (MPLNET), the Halocarbons and other Trace Species Network (HATS), the Southern Hemisphere Additional Ozonesonde Network (SHADOZ) and Total Carbon Column Observing Network (TCCON).

A fundamental aspect of NDACC is represented by the high quality standard of the collected data, which we demonstrate to be also reached by BASIL based on the results illustrated in this paper. Measurements of vertical profiles of atmospheric

temperature, water vapour mixing ratio and particle backscattering coefficient at 354.7 nm from BASIL are included in the NDACC database. BASIL is the only lidar system within the network which provides simultaneous and co-located measurements of these three atmospheric variables, with the data for these three variables being ingested in the NDACC repository and made available to the NDACC community.

BASIL in located in Potenza, Italy (40$^o$38'45" N, 15$^o$48'29" E, elevation: 730 m). The system is located in a sea-tainer on

the roof of Scuola di Ingegneria (main building) at Università degli Studi della Basilicata. The system includes a Nd:YAG laser, with both second and third harmonic generation crystals (average power: 10 W). BASIL uses a telescope in Newtonian configuration, with a 45 cm diameter primary mirror (f/2.1). BASIL performs accurate and high-resolution measurements of atmospheric water vapour and temperature, both in daytime and night-time, based on the exploitation of the vibrational and rotational Raman lidar techniques, respectively, in the ultraviolet (Whiteman, 2003; Di Girolamo et al., 2009,; Behrendt and

Reichardt, 2000; Di Girolamo et al., 2004, 2006, 2018a; Bhawar et al., 2011). BASIL also carries out measurements of the particle backscattering and extinction coefficient and depolarization at 354.7 nm. Relative humidity profiles are obtained from simultaneous water vapour mixing ratio and temperature profile measurements (Di Girolamo et al., 2009b). A transportable version of the system, emitting two additional wavelengths (523 and 1064 nm), has been deployed in a variety of international field experiments (Bhawar et al., 2008; Serio et al., 2008; Wulfmeyer et al., 2008; Bennett et al., 2011;

Ducrocq et al., 2014; Macke et al., 2017, Di Girolamo et al., 2012a, 2012b, 2016, 2017, 2018b). BASIL was included in NDACC with the primary aim of providing water vapour mixing ratio and temperature profile measurements. Thus, a major emphasis has been put in the collection and data processing for these variables, especially for what concerns the calibration and validation efforts. In the frame of NDACC, BASIL performs routine measurements each Thursday, typically from local noon to midnight couple of hours after sun set.

In addition to a larger accuracy and vertical resolution, a further advantage of lidar techniques with respect to traditional passive remote sensors is represented by the accurate characterization of the random uncertainty affecting the measurements, which is available for altitude and each individual profile. This is determined from the signal photon number based on the application of Poisson statistics. The application of Poisson statistics to lidar signals is a correct when dealing with lidar echoes acquired both in photon-counting and analogical mode. In this latter case analogical lidar signals must first be

converted into "virtual" counts. Considering an integration time of 5 min and a vertical resolution of 150 m, measurement precision at 10 km is typically 5% for water vapour mixing ratio and 1 K for temperature for night-time measurements. A detail description of the system setup has been provided in several previous publications (among others, Di Girolamo et al., 2009a, b).



### 3 Additional profiling sensors and model data involved in the inter-comparison effort

#### 3.1 IASI

The Infrared Atmospheric Sounding Interferometer (IASI), onboard the polar orbiting MetOp satellite series, is a nadir-viewing Fourier transform spectrometer measuring the Earth atmosphere emitted radiation in the thermal infrared region

$(3.2\text{-}15.5 \ \mu m$ or 645-2760 cm$^{-1}$), with an apodized spectral resolution of 0.5 cm$^{-1}$ (Collard, 2007; Masiello *et al.*, 2013). With at a horizontal resolution of 12 km over a swath width of 2200 km, IASI performs 14 sun-synchronous orbits with overpasses at 9:30 local time, ensuring global coverage twice per day. The main objective of IASI is to provide accurate and high resolution measurements of atmospheric temperature and humidity profiles. Temperature profiles are measured in the troposphere and stratosphere in clear-sky conditions, with an accuracy of 1 K and a vertical and horizontal resolution of 1

and 25 km, respectively, in the lower troposphere. Humidity profiles are measured in the troposphere under cloud-free conditions, with an accuracy of 10 % and a vertical and horizontal resolution of 1-2 and 25 km, respectively. Such performance has a major impact on many scientific areas, especially on Numerical Weather Prediction. IASI also provides measurements of trace gases concentrations, land and sea surface temperature and emissivity and cloud properties. For the purpose of this paper, we used the data product called IASI L2 TWT, available via EUMETCast, containing atmospheric

temperature and humidity profiles at 101 pressure levels and surface skin temperature. Profiles are provided at single IASI footprint resolution, with a horizontal resolution at nadir of about 25 km. The quality of the vertical profiles retrieved in cloudy IFOVs is strongly dependent on cloud properties available in the IASI CLP product and from co-located microwave measurements.

#### 3.2 AIRS

The Atmospheric Infrared Sounder (AIRS), launched aboard NASA's Aqua EOS satellite in 2002, is a hyper-spectral sensor including 2378 infrared channels and 4 visible/near-infrared channels, covering the spectral interval 3.7-15.4 μm (2665-650 cm$^{-1}$), with a spectral resolution $\lambda/\Delta\lambda$ of 1200. AIRS   is operated in combination with two microwave instruments, the Advanced Microwave Sounding Unit (AMSU-A) and the Humidity Sounder for Brazil (HSB), equipped with 15 and 4 microwave channels, respectively.

The combined use of this ensemble of sensors allows to provide global coverage, accurate and high resolution measurements of atmospheric temperature and humidity profiles. Temperature profiles are measured in the troposphere and stratosphere in clear-sky conditions, with an accuracy of 1 K and a horizontal resolution of 50 km. Vertical resolution is 1 and 4 km for tropospheric and stratospheric measurements, respectively. Tropospheric humidity profiles are measured under cloud-free conditions, with a vertical resolution of 2 km and an accuracy of 15 and 50 % in lower and upper troposphere, respectively.

The Aqua satellite is located on a sun-synchronous orbit, with a nominal altitude of 705 kilometers and an orbiting period of 98.8 minutes, corresponding to ~ 14.5 orbits per day. Overpasses are at 1:30 a.m. and 1:30 p.m. local time in descending and ascending orbits, respectively. As for IASI, AIRS provides concentration measurements for a variety of trace gases. For the



purpose of this paper, we used the AIRS Version 6 Level 2 Standard Retrieval Product, which is based on 6-min data averaging (Boylan *et al.*, 2015).

**3.4 ECMWF**

Reanalysis from the European Centre for Medium-range Weather Forecasts (ECMWF) are also considered in this inter-

comparison effort. Two distinct reanalysis products are considered: ERA-15 (ECMWF, 2006), covering the 15-year period from December 1978 to February 1994, hereafter referred to as ECMWF, and ERA-40 (Uppala *et al.*, 2005), hereafter referred to as ECMWF-ERA40, originally intended to cover a 40-year period, but finally including a 45-year period from 1957 (International Geophysical Year) to 2002. This latter reanalysis makes use of a larger ensemble of archived data, which were not available at the time of the original analyses. Horizontal resolution of the data set is ~ 80 km, covering 60 vertical

levels from surface up to 0.1 hPa.

**4 Lidar measurements of atmospheric thermodynamic variables**

**4.1 Water vapour mixing ratio**

Raman lidar measurements of the water vapour mixing ratio profile have been extensively reported in the literature (Whiteman *et al.*, 1992; Whiteman, 2003). The approach makes use of the roto-vibrational Raman lidar signals from water

vapour and nitrogen molecules at the two Raman-shifted wavelengths $\lambda_{H_2O}$ and $\lambda_{N_2}$, respectively. When stimulating Raman scattering with ultraviolet laser light at wavelength $\lambda_0$=354.7 nm (i.e. the tripled frequency of the Nd:YAG laser source), $\lambda_{H_2O}$ and $\lambda_{N_2}$ are found to be located at 407.5 and 386.7 nm, respectively. These signals, expressed as number of detected photons from a given altitude $z$ above station level, are given by the expressions:

$$P_{H_2O}(z) = P_0 \frac{c \Delta t}{2} \frac{A_{tel}}{R^2} \eta_{H_2O} O(z) n_{H_2O}(z) \sigma_{H_2O} T_{\lambda_0}(z) T_{\lambda_{H_2O}}(z) \tag{1}$$

$$P_{N_2}(z) = P_0 \frac{c \Delta t}{2} \frac{A_{tel}}{R^2} \eta_{N_2} O(z) n_{N_2}(z) \sigma_{N_2} T_{\lambda_0}(z) T_{\lambda_{N_2}}(z) \tag{2}$$

where $P_0$ is the number of transmitted photons of each laser pulse at wavelength $\lambda_0$, $c$ is the speed of light, $A_{tel}$ is the telescope aperture area, $\eta_{H_2O/N_2}$ is the overall transmitter–receiver efficiency (inclusive of the reflectivity of the telescope primary-secondary mirrors and the transmission optics, the interference filter transmission and the detector quantum efficiency) at wavelength $\lambda_{H_2O}/\lambda_{N_2}$, $\Delta t$ is the laser pulse duration, $n_{H_2O}(z)/n_{N_2}(z)$ represents the water vapour/molecular

nitrogen number density, $\sigma_{H_2O}/\sigma_{H_2O}$ is the water vapour/molecular nitrogen roto-vibrational Raman cross-section, $T_{\lambda_0}(z)$ and $T_{\lambda_{H_2O}}(z)/T_{\lambda_{N_2}}(z)$ are the atmospheric transmission profiles from surface up to the scattering volume altitude $z$ at $\lambda_0$ and $\lambda_{H_2O}/\lambda_{N_2}$, respectively. The water vapour mixing ratio profile, $x_{H_2O}(z)$, can be determined from the power ratio of $P_{H_2O}(z)$ and $P_{N_2}(z)$ through the expression:





$$x_{H_2O}(z) = K(z) \cdot \frac{P_{H_2O}(z)}{P_{N_2}(z)} \qquad (3)$$

The calibration function $K(z)$ is determined through a calibration procedure, which is described in detail in Di Girolamo *et al.* (2018a), based on the comparison between simultaneous and co-located water vapour mixing ratio profiles from the lidar and an independent humidity sensor. For the purpose of this study, the estimate of $K(z)$ is based on an extensive comparison
between BASIL and the radiosonde data from the nearby station CIAO.

**4.2 Temperature**

In the recent past, temperature lidar measurements have become more and more important in weather and climate studies. Several lidar techniques have demonstrated to be effective for routine measurements (Behrendt, 2005). Among others, the rotational Raman technique (Behrendt and Reichardt, 2000) and the integration technique (Hauchecorne and Chanin, 1980;
Hauchercorne *et al.*, 1992). The rotational Raman technique, especially if implemented in the UV, allows measuring temperature profiles typically up to the lower stratosphere, while the integration technique is successfully used to measure temperature profiles throughout the stratosphere and mesosphere.
An effective use of the integration technique implies the use of very powerful laser sources in combination with large aperture telescopes and complex receiving systems. Receivers are typically equipped with mechanical or electro-optical
choppers in order to prevent detection of elastic echoes from lower levels, primarily the troposphere, which may overload detectors and induce non-linear responses (signal-induced noise) in the upper level signals, typically those collected from the stratosphere and mesosphere (Di Girolamo *et al.*, 1994). The use of mechanical choppers, usually located just below the telescope focus (Sica *et al.*, 1995), imposes the implementation of separate lidar receivers for the purpose of achieving a successful simultaneous exploitation of the rotational Raman and the integration techniques. A simpler optical design
solution can be considered in case of use of a moderate power laser source and a smaller aperture telescope. In this regard, it is to be pointed out that performing accurate temperature measurements through the integration lidar technique imposes the use of lidar systems with large values of the power-aperture (PA) product. Values of the PA product for temperature lidars exploiting the integration technique are usually in excess of 10 $Wm^2$ (Hauchercorne *et al.*, 1992), with values for specific systems in excess of 50 $Wm^2$ (Sica *et al.*, 1995). The Raman lidar system considered in the present paper is characterized by
a PA product not exceeding 1 $Wm^2$. Such a low PA product value allows performing temperature profile measurements based on the simultaneous application of the rotational Raman and the integration techniques, without the need of complex receiving systems. Such low PA product value allows simultaneous temperature measurements by both the rotational Raman technique, up to approx. 25 km, and the integration technique, from 20 km up to approx. 50 km, with no contamination of the elastic signals by signal-induced noise effects. To the best of our knowledge, these measurements represent the first
successful demonstration of the simultaneous application in a single instrument of both the rotational Raman and integration lidar techniques in the ultraviolet spectral region, i.e. in the region where the simultaneous exploitation of these two techniques has the highest potential.




### 4.2.1 Rotational Raman technique

Rotational Raman lidar measurements of the atmospheric temperature profile rely on the use of the rotational Raman backscattered signals from nitrogen and oxygen molecules within two narrow spectral regions encompassing rotational lines from these two species with opposite sensitivity to temperature changes: rotational lines which are closer to the laser

wavelength $\lambda_0$, characterized by lower values of the rotational quantum number $J$, increase in intensity with decreasing temperature, while rotational lines which are distant from the laser wavelength, characterized by higher values of $J$, show the opposite behavior, with their intensity increasing with increasing temperature.

Atmospheric temperature measurements are obtained from the ratio of the signal including low quantum number $J$ rotational lines, $P_{LoJ}(z)$, over the signal including high quantum number $J$ rotational lines, $P_{HiJ}(z)$, with centre wavelengths being $\lambda_{LoJ}$

and $\lambda_{HiJ}$, respectively. Specifically, the atmospheric temperature profile, $T(z)$, is obtained from the signal ratio $R(T)=$ $P_{HiJ}(z)/P_{LoJ}(z)$, through the inversion of the following expression:

$$R(z) = \frac{P_{HiJ}\ (z[T])}{P_{LoJ}\ (z[T])} \cong \exp(\frac{a}{T(z)} + b) \qquad (4)$$

where $a$ and $b$ are two calibration constants, which can be determined based on the comparison of Raman lidar measurements with simultaneous and co-located temperature measurements. Thus, $T(z)$ is obtained through the analytical

expression:

$$T(z) = \frac{a}{\ln R(z) - b} \qquad (5)$$

The location of the rotational Raman signals center wavelengths $\lambda_{LoJ}$ and $\lambda_{HiJ}$ was determined through a specific sensitivity study accounting for the temperature sensitivity of rotational lines' intensity and the variable solar background conditions (Hammann and Behrendt, 2015). In the definition of the properties of the spectral selection devises (interference filters), $\lambda_{LoJ}$

and $\lambda_{HiJ}$ were selected with the purpose to guarantee comparable performance in daytime and nighttime and maximize measurement precision in the temperature range which is typically found throughout the troposphere (Di Girolamo *et al.*, 2004). Based on this selection, when using an ultraviolet laser wavelength at $\lambda_0=354.7$ nm, $\lambda_{LoJ}$ and $\lambda_{HiJ}$ are located at 354.3 and 352.9 nm, respectively.

### 4.2.2 Lidar integration technique

The  atmospheric number density profile, $N(z)$, can be determined from the elastic backscatter signal at wavelength $\lambda_0$, $P_{\lambda_0}(z)$, through the expression:

$$N(z) = \frac{C P_{\lambda_0}\ (z)z^2}{T_{\lambda_0}^2\ (z)} \qquad (6)$$



In this expression, $z$ represents the altitude above the station level, $T_{\lambda_0}(z)$ is the atmospheric transmission profile at $\lambda_0$ from surface up to the scattering volume altitude $z$, $C$ is a calibration constant including all instrumental parameters (laser power, telescope area, the signal transmission/reception efficiency, detector's quantum efficiency, etc.) and atmospheric parameters which are constant with altitude throughout the measurement integration time (aerosol and clouds below 20 km, see more

detail below).

Expression (6) assumes aerosol and clouds contributions to both $P_{\lambda_0}(z)$ and $T_{\lambda_0}(z)$ to be negligible; consequently, it can be assumed valid only in aerosol and cloud free atmospheric regions. This implies that $N(z)$ can be estimated from the elastic backscatter signal $P_{\lambda_0}(z)$ through expression (6) only in unperturbed stratospheric conditions at altitudes above 30 km, i.e. above the background stratospheric aerosol occasionally observed in the lower stratosphere. The background stratospheric

aerosol layer typically consists of concentrated solutions of sulfur acid, which is produced in chemical reactions involving sulfur dioxide transported from the troposphere (Rosen, 1971; Ugolnikov and Maslov, 2018). An accurate determination of the calibration constant $C$ requires the availability of ancillary information. A possible approach to quantify $C$ considers the normalization of the quantity $P_{\lambda_0}(z)z^2/T_{\lambda_0}^2(z)$ to a profile of $N(z)$ from either a climatological atmospheric model or a co-located simultaneous radiosonde. The normalization is carried out in the aerosol and cloud free atmospheric region, i.e.

above 20 km, over a vertical region including the reference altitude $z_{ref,1}$ through the expression:

$$C = \frac{\sum_{i=-k}^{+k} \dfrac{N(z_{ref,1+i})T_{\lambda_0}^2(z_{ref,1+i})}{P_{\lambda_0}(z_{ref,1+i})z_{ref,1+i}^2}}{2k+1}$$

(7)

with $T_{\lambda_0}(z_{ref,1+i})$ being the atmospheric transmission profile at $\lambda_0$ from surface level up to the scattering volume at altitude $z_{ref,1+i}$. A number $k$ of altitude levels below and above the reference altitude $z_{ref,1}$ are considered in order to reduce the random uncertainty affecting the estimate of $C$. For example, when considering a vertical lidar resolution of 150 m and an aerosol

and cloud free atmospheric region for normalization of 1500 m, $k$ varies from -5 to +5.

Once $N(z)$ is determined, the temperature profile can easily be derived. For this purpose we consider the ideal-gas law in the form:

$$p(z) = k\, N(z)\, T(z)$$

(8)

with $p(z)$ being the atmospheric pressure profile, $T(z)$ being the atmospheric temperature profile and $k$ being the Boltzmann

constant ($1.38 \times 10^{-23}$ J K$^{-1}$). We also consider the barometric altitude equation, also known as hydrostatic equation, which can be expressed in the form:

$$\mathrm{d}p(z) = -\rho(z)\, g(z)\, \mathrm{d}z$$

(9)





where $\rho(z)$ is the atmospheric mass density profile and $g(z)$ is the gravitational acceleration. Equation (9) is valid under hydrostatic equilibrium conditions. The combination of equations (8) and (9) leads to the following expression (Hauchecorne and Chanin, 1980):

$$T(z) = \frac{N(z_{ref,2})}{N(z)} T(z_{ref,2}) + \frac{M}{kN(z)} \int_{z_{ref,2}}^{z} g(\varsigma) N(\varsigma) d\varsigma \tag{10}$$

where the atmospheric mass density profile has been expressed as $\rho(z) = M \times N(z)$, with $M$ being the apparent molecular weight of atmosphere (28.97), which is considered to be constant throughout the homosphere (up to 100 km).

This algorithm can be applied starting from a reference maximum altitude, hereafter identified with the symbol $z_{ref,2}$, assuming to know the atmospheric number density and temperature values at this altitude, i.e. $N(z_{ref,2})$ and $T(z_{ref,2})$. Imposing these boundary conditions, $T(z)$ can be derived starting from the reference altitude $z_{ref,2}$ and progressively

extrapolating the algorithm down to lower levels. Temperature at an altitude $z_{ref,2+1}$, immediately below $z_{ref,2}$, can be expressed as:

$$T(z_{ref,2+1}) = \frac{N(z_{ref,2})}{N(z_{ref,2+1})} T(z_{ref,2}) + \frac{M}{kN(z_{ref,2+1})} g_{med} N_{med} \Delta z \tag{11}$$

with $\Delta z = z_{ref,2+1} - z_{ref,2}$ and $g_{med}$ and $N_{med}$ being the mean gravitational acceleration and atmospheric number density, respectively, between $z_{ref,2}$ and $z_{ref,2+1}$. These can be expressed as (Behrendt, 2005):

$$g_{med} = \frac{g(z_{ref,2}) + g(z_{ref,2+1})}{2} \tag{12}$$

and

$$N_{med} = \frac{N(z_{ref,2}) - N(z_{ref,2+1})}{\ln \frac{N(z_{ref,2})}{N(z_{ref,2+1})}} \tag{13}$$

The algorithm can be applied both in the downward and upward direction. Consequently the reference altitude $z_{ref,2}$ can be

taken at the highest or lowest boundary level of the vertical region where the integration technique for is applied (Behrendt, 2005). However, boundary values $T(z_{ref,2})$ and $N(z_{ref,2})$ must be known with sufficiently high accuracy if temperature profiles are to be extrapolated upward because errors build up exponentially when proceeding in this direction (Behrendt, 2005). On the contrary, when an upper reference altitude is taken and the algorithm is applied  downward, errors affecting $T(z_{ref,2})$ and $N(z_{ref,2})$ quickly reduce (Behrendt, 2005). This is the motivation why most lidar groups, including us, usually apply this

algorithm downward, typically considering values of $T(z_{ref,2})$ and $N(z_{ref,2})$ from atmospheric climatological models or satellite data (Behrendt, 2005). Systematic errors associated with an incorrect selection of $T(z_{ref,2})$ and $N(z_{ref,2})$in the downward integration of the algorithm given in expression (11) were investigated by Leblanc *et al.* (1998), considering a value for $z_{ref,2}$=90 km and using, as a worst-case scenario, a reference value for $T(z_{ref,2})$ exceeding by 15 K the corresponding model





value at this same altitude. Leblanc *et al.* (1998) revealed that the bias was already reduced to 4 K at 80 km and to 1 K at 70 km. In real measurements, the considered value for $T(z_{ref,2})$ is expected to be much closer to its correct value. Consequently, systematic errors in the temperature profile associated with the selection of wrong temperature boundary conditions and the application of the downward-integration technique are very small (~ 1 K, Behrendt, 2005). It is to be specific that only

systematic errors associated with the selection of a wrong value of $T(z_{ref,2})$ are to be considered, while those associated with the selection of a wrong value of $N(z_{ref,2})$ are always negligible because deviations of real atmospheric number density profiles from climatological profiles are always very small (1-2 %) in the altitude region where boundary conditions are typically selected (50-90 km).

**4.3 Relative humidity**

The availability of simultaneous and co-located measurements of the water vapour mixing ratio and temperature profiles, as is the case for BASIL, makes the determination of the relative humidity profile straightforward. Relative humidity (with respect to water) is defined as the ratio, expressed in percentage, between the water vapour partial pressure profile $e(z)$ and the saturated vapor pressure profile $e_{sat}(z)$, i.e. $RH(z) = 100 \times e(z)/e_{sat}(z)$. $e(z)$ can be expressed as:

$$e(z) = \frac{p(z)x_{H_2O}(z)}{0.622 + x_{H_2O}(z)} \qquad (14)$$

with $p(z)$ being the atmospheric pressure profile, usually taken from simultaneous measurements with other sensors (for example radiosondes) or obtained from surface pressure measurements, assuming hydrostatic equilibrium and applying the hydrostatic equilibrium equation. A commonly used expression for $e_{sat}(z)$ (List, 1951) is given by:

$$e_s(z) = 6.108 \exp\left\{\frac{17.08[T(z) - 273.15]}{T(z) - 38.97}\right\} \qquad (15)$$

with $T(z)$ being expressed in degrees Celsius. As $e_{sat}(z)$ depends is only on $T(z)$, $RH(z)$ can be determined directly from

BASIL measurements of $x_{H_2O}(z)$ and $T(z)$, based on the only knowledge of the surface pressure value.

**5 Statistical quantities used for the inter-comparison**

In order to assess the performance of the different profiling sensors and models considered in the study, an appropriate statistical analysis has to be carried out based on the estimation of specific statistical quantities. Specifically, for each sensor/model pair, the relative bias and root-mean square deviation profile between two profiles, can be determined through

the following expressions (Behrendt *et al.*, 2007a, 2007b, Bhawar *et al.*, 2011):

$$BIAS = \frac{1}{N}\sum_{i=1}^{N} BIAS_i = \frac{2}{N}\sum_{i=1}^{N}\left\{\frac{\sum_{z=z_1}^{z_2}[q_1(z) - q_2(z)]}{\sum_{z=z_1}^{z_2}[q_1(z) + q_2(z)]}\right\} \qquad (16)$$


$$RMS = \frac{1}{N}\sum_{i=1}^{N} RMS_i = \frac{2}{N}\sum_{i=1}^{N}\left\{ \frac{\sqrt{N\sum_{z=z_1}^{z_2}[q_1(z)-q_2(z)]}}{\sum_{z=z_1}^{z_2}[q_1(z)+q_2(z)]} \right\}$$

(17)

where $q_1(z)$ where $q_2(z)$ represent the water vapor mixing ratio or temperature values at altitude z for sensor/model 1 and sensor/model 2, respectively, $z_1$ and $z_2$ are the lower and upper levels of the considered altitude interval, respectively, and $N$ is the number of data points for each sensor/model in this interval. In the expressions above we used the mean of the

measurement result of the two sensors/models as reference instead of using the measurement result of one of the two. This approach leads to more objective results than considering one of the sensors/models as reference (Behrendt *et al.* (2007a,b). For all inter-comparisons reported in this paper, bias and RMS deviation are computed in the 500 m altitude intervals ($z_2$-$z_1$=500 m). The index *i*, having values in the range from 1 to $N$, identify the inter-comparison sample, where $N$ is the total number of possible comparisons for each sensor/model pair. Profiles of mean bias and RMS deviation are finally computed

taking into consideration the total number of possible inter-comparisons for each sensor/model pair. For the purpose of applying expressions (16) and (17) we considered a common altitude array for each pair of sensors. Consequently, in case of different altitude arrays for the compared profiles, data from one sensor/model have to be interpolated to the other sensor/model altitude levels. The absolute bias and root-mean-square deviation can be determined from expressions (16) and (17), respectively, through their multiplication by the mean of the two profiles:

$$\frac{2\sum_{z=z_1}^{z_2}\{q_1(z)+q_2(z)\}}{N}$$
(18)

The estimate of the bias and root-mean square deviation between two compared sensors/models allows quantifying the mutual performance of the two , i.e. how one performances with respect to the other. The bias, which quantifies the relative accuracy of the compared sensors/models, identifies an offset between the two, which is attributable to different sources of systematic uncertainty affecting one or both sensors/models. As opposed to this, the root-mean-square deviation includes all

possible differences between the two sensors/models, associated with both systematic and statistical uncertainties and with changes of the measured/modeled atmospheric parameter (water vapour mixing ratio or temperature) as a result of differences in the considered air masses. Based on expressions (16) and (18), the absolute and percentage bias of the sensor/model 1 vs. the sensor/model 2 has positive values when $q_1(z)$ is larger than $q_2(z)$, i.e. $q_1(z)$ overestimates $q_2(z)$ or $q_2(z)$ overestimates $q_1(z)$.

**6 Inter-comparison results**

BASIL was approved to enter NDACC in November 2012 and started operations shortly afterwards. However, routine measurements on a weekly basis started only one year later. In this paper we report measurements performed during the two year period from 7 November 2013 to 5 October 2015. During this time interval BASIL collected 385 hours of



measurements distributed over 80 days. Lidar measurements are compared with model re-analyses (ECMWF, ECMWF-ERA40 ), satellite data (IASI, AIRS), and radiosondes from CNR in Tito. Figure 1 shows the location of BASIL (Lat: 40,60°N, Long.: 14,85 °E), together with the footprint of AIRS (centered at Lat: 40,50 °N and Long: 15,50 °E, size: 72×72 km) and IASI (centered at Lat: 40,89 °N and Long: 16,02 °E, size: 12×12 km) and the size of grid point of ECMWF ERA-

15 and ECMWF ERA-40 (centered at Lat: 40,63 °N and Long: 15,75 °E size: 9×9 km). The distance between BASIL and the center point of the other sensors/models is variable, i.e. 8 km with CIAO radiosonde launching facility at IMAA-CNR, 25 km with AIRS, 25 km with IASI and 4 km with ECMWF ERA-15 and ECMWF ERA-40.

For the purpose of this study we selected four case studies, covering different measurement and meteorological conditions.

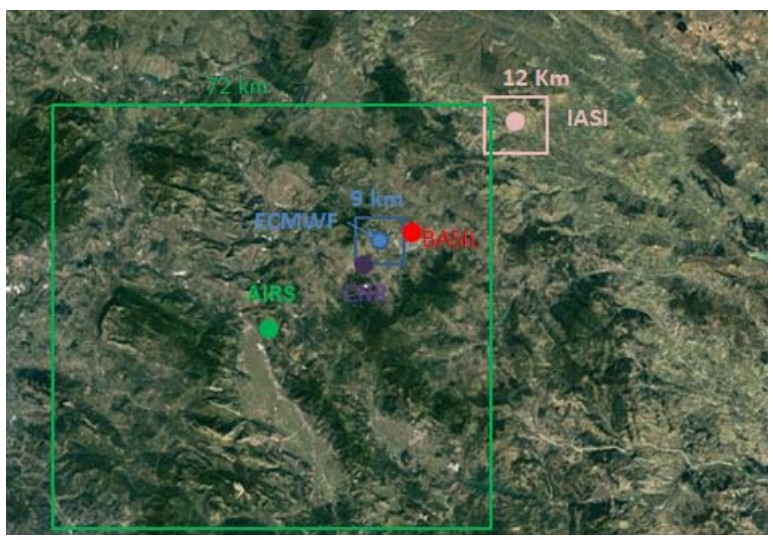

Figure 1: Location of BASIL (red dot), together with the footprint of AIRS (green square centered on green dot) and IASI (pink square centered on pink dot), CIAO radiosonde launching facility (purple dot) and the grid point of ECMWF ERA-15 and ECMWF ERA-40 (blue square centered on blue dot).

**6.1 Case study on 7 November 2013**

For the aims of this paper, we focused our attention on four selected case studies collected during the first 2 years of
operation of the system, namely 7 November 2013, 9 October 2014 and 2 and 9 April 2015. While a larger data-set could have been chosen, we decided to focus our attention to a limited number of case studies, which have been carefully analyzed with a customized approach, instead of considering a larger dataset analyzed with a standard routine analysis approach. Such an approach was considered with the purpose of minimizing the effects associated with the application of the data analysis





approach on the results of the performed statistical analysis. The four selected case studies cover a two year time period strating from the early stage of BASIL operation in the frame of NDACC (7 November 2013 to April 2015).

Figure 2a illustrates the mean water vapour mixing ratio profile measured by BASIL on 7 November 2013 over the time interval 17:00-19:00 UTC. The vertical resolution of the data is 150 m from surface up to 6 km, 300 m between 6 and 8 km

and 600 m above 8 km. The water vapour mixing ratio profile from BASIL reaches an altitude of approx. 15 km, with the capability to measure humidity levels as small as 0.003-0.004 g kg$^{-1}$, with a sensitivity level of 0.001-0.002 g kg$^{-1}$. The capability to reach an altitude of 14-15 km, with a measurement detection level of 0.001-0.002 g kg$^{-1}$, has been verified in most of the 2 h water vapour profiles measured in the frame of NDACC in clear sky conditions by BASIL. When considering measurements integrated in time over the entire night, the water vapour mixing ratio profile by BASIL is found

to extend up to approx. 16-18 km. For this case study, the closest in time the water vapour mixing ratio profiles from IASI (at 19:29 UTC) and AIRS (at 14:09 U.T.C.) and the model re-analysis ECMWF ERA-15 and ECMWF ERA-40 (at 18:00 UTC) are also illustrated in figure 2a. In the present case study, no radiosonde was launched from the nearby CIAO launching station. The agreement between BASIL and the different sensors/models is very good, even at low altitudes where effects of water vapour heterogeneity are usually important. For this specific case study, at all altitudes above 2.5 up to 15

km, deviations of BASIL vs. AIRS and IASI are smaller than 0.2 g kg$^{-1}$ (or 40 %) and 0.1 g kg$^{-1}$ (or 30 %), respectively, while deviations between BASIL and ECMWF (ERA-15 and ERA-40) are not exceeding 0.1 g kg$^{-1}$ (or 30 %). The mean bias of BASIL vs. AIRS, IASI, ECMWF and ECMWF -ERA40 are -16 % (or -0.13 g kg$^{-1}$), 7% (or 0.08 g kg$^{-1}$), -5% (or 0.0032g kg$^{-1}$) and -11% (or -0.05 g kg$^{-1}$), respectively.

Figure 2 b shows the time evolution of the water vapour mixing ratio over a 6 h time interval from 16:00 to 22:00 UTC on

this same day (7 November 2013). The figure is a succession of 72 consecutive 5-min averaged profiles. For the purpose of reducing signal statistical fluctuations, a vertical smoothing filter was applied to the data, finally achieving an overall vertical resolution of 150 m. Figure 2 b reveals the presence of a well-defined humid layer extending from surface up to ~1.5 km between 16:00 and 17:00 UTC and then progressively reducing in depth down to 2-300 m, which identifies the evolution of the convective boundary-layer in its final decaying phase at the solar portion of the day. A variety of humidity layers are

visible above. The ability to perform water vapour mixing ratio profile measurements with such high time resolution is a unique feature of Raman lidars.





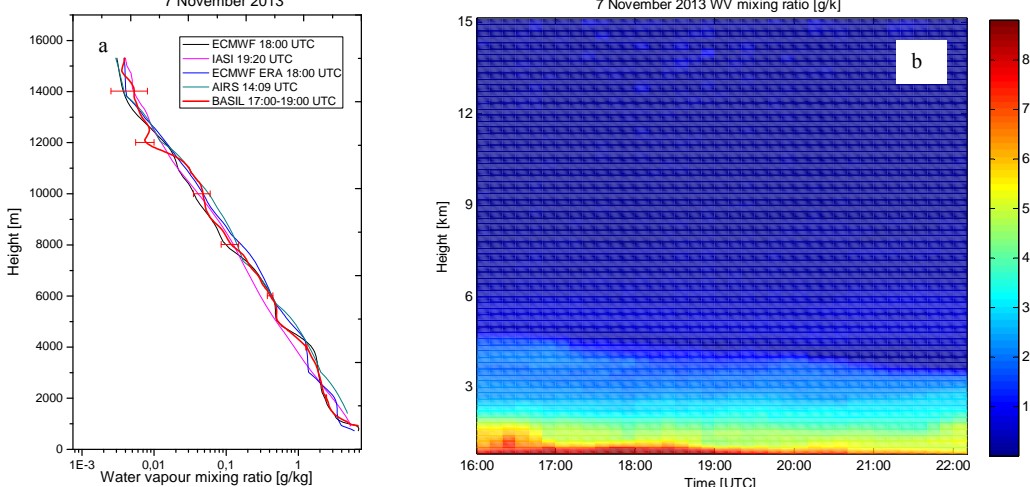

Figure 2: Water vapour mixing ratio profile as measured by BASIL over the time period 17:00-19:00 UTC on 7 November 2013, together with the closest in time profiles from IASI (at 19:20 UTC), AIRS (at 14:09 UTC) and the model re-analysis ECMWF (ERA-15 and ERA-40, at 18:00 UTC) (panel a). Time evolution of water vapour mixing ratio profile as measured by BASIL over the interval 16:00-22:00 UTC on 7 November 2013 (panel b).

Figure 3a illustrates the mean atmospheric temperature profile measured by BASIL on 7 November 2013 over the same time interval considered in figure 2a. The measurement is based on the use of the rotational technique up to 20 km and the integration technique above 20 km. The combined use of these two techniques allows temperature profile measurements up to 50-55 km. In the altitude region where the rotational Raman technique is exploited the vertical resolution of the data is 150 m from surface up to 6 km and 600 m above this altitude. The integration technique is applied downward, initializing the algorithm at 55 km. As mentioned above, although the boundary value of $T(z_{ref,2})$ taken from a model atmosphere may differ from its real value, the systematic error affecting the measurement becomes negligible 5-7.5 km below this level (Hauchercorne *et al.*, 1992). For this motivation, profiles in figure 3a and b are shown only below 50 km.

Again, the closest in time temperature profiles from the sensors IASI (at 19:20 UTC) and AIRS (at 14:09 UTC) and the model re-analysis ECMWF and ECMWF-ERA40 (at 18:00 UTC) are also illustrated in figure 3a. The agreement between BASIL and the different sensors/models is very good. Specifically, deviations between BASIL and AIRS/IASI are smaller than 2 K from surface up to 40 km and smaller than 3-5 K above. Deviations between BASIL and ECMWF analysis (ERA-15 and ERA-40) are not exceeding 2 K all the way up to 50 km. It is to be pointed out that deviations between BASIL and the other sensors/models may be the results of the random and systematic uncertainties affecting the different sensors, as well as of the different air masses sounded by the different sensors or encompassed in the different grid points. However, it



is to be added that temperature measurements by lidar frequently reveal temperature fluctuations associated with the propagation of internal gravity waves (Di Girolamo *et al.*, 2009a). These fluctuations, having amplitudes increasing with increasing altitude, can be as large as 5-15 K (Chanin *et al.*, 1994; Zhao *et al.*, 2017). Consequently, deviations between BASIL and the other sensors/models are in part possibly associated with the effects of gravity waves. The mean bias of

BASIL vs. AIRS, IASI, ECMWF and ECMWF-ERA40 are 1.05 K, 0.83 K, 0.41 K and -0.72K, respectively.

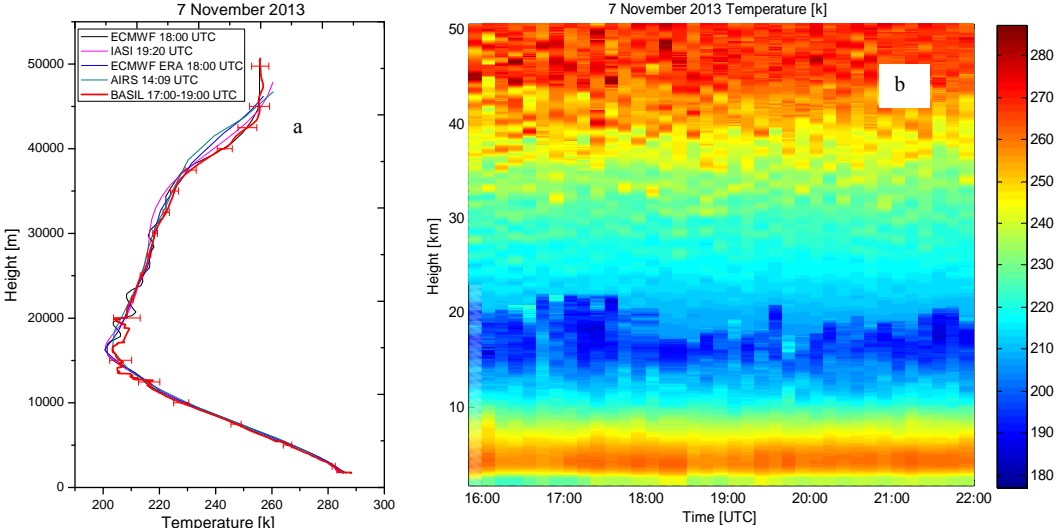

Figure 3: Vertical profile of atmospheric temperature as measured by BASIL over the time period 17:00-19:00 UTC on 7 November 2013, together with the closest in time profiles from IASI (at 19:20 UTC), AIRS (at 14:09 UTC) and the model re-analysis ECMWF (ERA-15 and ERA-40, at 18:00 UTC) (panel a). Time evolution of atmospheric temperature as

measured by BASIL over the interval 18:00-22:00 UTC on 7 November 2013 (panel b).

Figure 3b shows the evolution of atmospheric temperature over the same 6 h time interval considered in figure 2b. Again, the figure is a succession of 72 consecutive 5-min averaged profiles. In this case, for the purpose of obtaining a sufficiently high signal statistics, a vertical resolution of 150 m was considered. It is to be noticed that, despite the short integration time, the strong signal intensities in combination with favourable clear weather conditions allows reaching an altitude of 50 km. The

top of the convective boundary layer is clearly visible in the figure, this being identified by the strong temperature gradient around 1.5-2 km. The tropopause region and its fluctuations are also clearly visible in the figure.

Accurate relative humidity (RH) measurements are of paramount importance to determine cloud and aerosol radiative properties and related microphysical processes. RH has been demonstrated to have a critical influence on aerosol climate forcing (Pilins *et al.*, 1995). Aerosol hygroscopic growth at high relative humidity values may significantly influence aerosol

direct effect on climate (Wulfmeyer and Feingold, 2000). As described in section 4.3, RH profiles are obtained from the





simultaneous and independent measurements of the water vapour mixing ratio and temperature profiles carried out by BASIL. Figure 4a illustrates the mean atmospheric relative humidity profile measured by BASIL on 7 November 2013 over the same time period considered in figures 2a and 3a. The agreement between BASIL and the different sensors/models is good, with deviations not exceeding 10 % up to 15 km.

Figure 4b shows the time evolution of relative humidity over the same 6 h interval considered in figure 2b and 3b, the figure being again a succession of 72 consecutive 5-min averaged profiles with a vertical resolution of 150 m. It is to be noticed that, despite the short integration time, an altitude of 15 km is reached, with measurements revealing a variability of RH in the UTLS region systematically larger than the random uncertainty affecting the Raman lidar measurements.

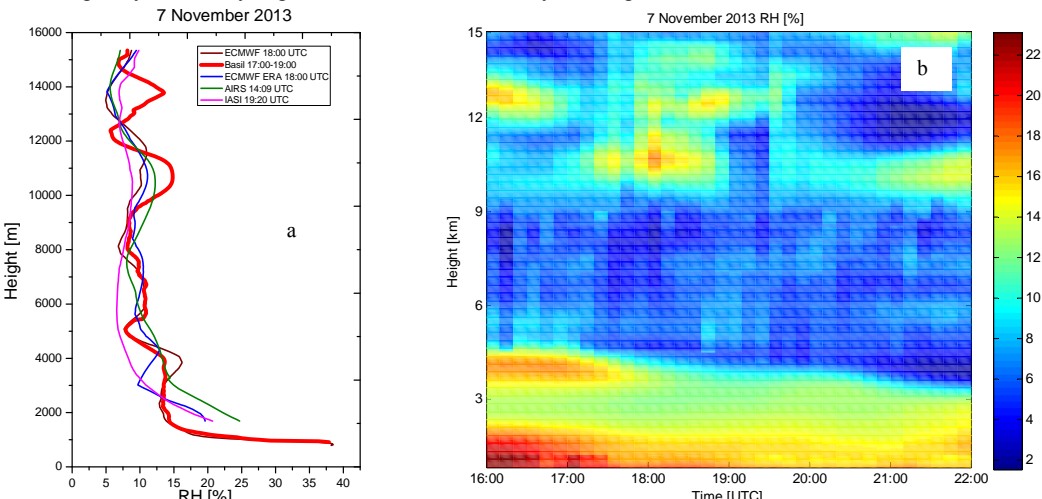

Figure 4: Vertical profile of relative humidity as measured by BASIL over the time period 17:00-19:00 UTC on 7 November 2013, together with the closest in time profiles from IASI (at 19:20 UTC), AIRS (at 14:09 UTC) and the model re-analysis ECMWF (ERA-15 and ERA-40, at 18:00 UTC) (panel a). Time evolution of relative humidity as measured by BASIL over the interval 18:00-22:00 UTC on 7 November 2013 (panel b).

**6.2 Other case studies: 9 October 2014, 2 and 9 April 2015**

Figure 5a illustrates the mean water vapour mixing ratio profile measured by BASIL on 9 October 2014 over the time interval 16:00-18:00 UTC, together with the closest in time water vapour mixing ratio profiles from IASI (at 19:20 UTC) and AIRS (at 14:35 UTC) and the model re-analysis ECMWF ERA-15 and ECMWF ERA-40 (at 18:00 UTC). The vertical resolution of the Raman lidar data is 150 m from surface up to 6 km, 300 m between 6 and 8 km and 600 m above 8 km, with water vapour mixing ratio found to reach an altitude of approx. 14 km, revealing values as small as 0.02 g kg$^{-1}$. The

present comparison also includes the profile from the radiosonde launched at 18:00 UTC on this same day from the nearby station IMAA-CNR (~ 7 km W), which was not available for the previous case study. Deviations between BASIL and the





radiosonde is not exceeding 0.1 g kg$^{-1}$ at all altitudes up to ~ 9 km. The mean bias between BASIL and the radiosonde  is 11% (or 0.03 g kg$^{-1}$), while the mean bias of BASIL vs. AIRS, IASI, ECMWF and ECMWF-ERA40 are  44 % (or 0.39 g kg$^{-1}$),  30% (or 0.26 g kg$^{-1}$),  44% (or 0.39 g kg$^{-1}$) and 17% (or 0.29 g kg$^{-1}$),  respectively. Figure 5b illustrates the mean atmospheric temperature profile measured by BASIL over the same time period considered in figure 5a, together with the

corresponding profiles from IASI, AIRS and the model re-analysis (ECMWF ERA-15 and ECMWF ERA-40). Again, the lidar measurement is based on the use of the rotational technique up to 20 km and the integration technique above. As also argued for the previous case study, deviations between BASIL and the other sensors/models observed above 35 km are possibly associated with the effect of gravity waves propagation, whose presence is revealed by the Raman lidar and missed by the other sensors/models. The temperature profile measured by the radiosonde extends up to 30 km. The mean bias

between BASIL and the radiosonde is  0.58 K, while the mean bias of BASIL vs. AIRS, IASI, ECMWF and ECMWF-ERA40 are 2.13 K, 0.88 K, -0.68 K and 0.74 K, respectively. Finally, figure 5c illustrates the mean relative humidity profiles measured by BASIL and the other sensors/models over the same time intervals considered in figure 5a and b, with deviations between BASIL and the radiosonde being minimum in the free troposphere and being maximum in the atmospheric boundary layer (ABL) and the UTLS region. Water vapour heterogeneity is larger in the ABL than in the free troposphere because of the larger effects of surface sources and sinks on water vapour distribution. As a result of atmospheric

heterogeneity, appreciable differences in moisture content can be observed in the ABL because of the distance (~7 km) between the lidar facility (within the urban area of Potenza) and the radiosonde launching facility (countryside location). It is to be mentioned that the radiosonde may be horizontally drifted by wind during its vertical ascent. As a result of this drift, the radiosonde, when passing through the UTLS region, may have moved 1-200 km from the lidar vertical. Consequently,

even if the humidity field heterogeneity in the UTLS region is much lower than at lower altitudes, deviations between BASIL and the radiosonde observed in the UTLS region  may be associated with different air masses sounded by the two sensors in this region in addition to the large statistical uncertainty affecting Raman lidar measurements at these altitudes.





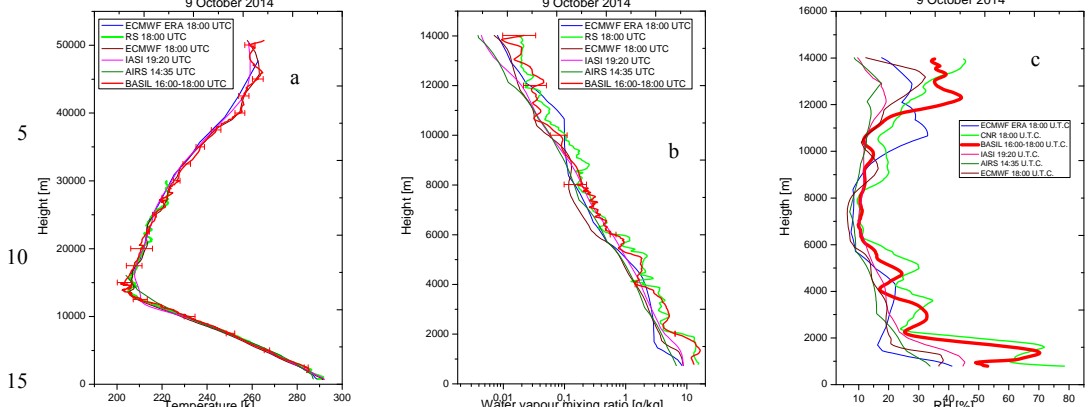

Figure 5: Mean profile of water vapour mixing ratio (a), temperature (b) and relative humidity (c) as measured by BASIL over the time period 16:00-18:00 UTC on 9 October 2014, together with the closest in time profiles from IASI (at 19:20 UTC), AIRS (at 14:35 UTC) and the model re-analysis ECMWF (ERA-15 and ERA-40, at 18:00 UTC).

Figure 6 illustrates the mean water vapour mixing ratio (a), temperature (b) and relative humidity (c) profile measured by BASIL on 2 April 2015 over the time interval 18:00-20:00 UTC, together with the simultaneous radiosonde profile (launched at 18:00 UTC) and the closest in time vertical profiles the water vapour mixing ratio profiles from IASI (at 18:59 UTC) and AIRS (at 15:18 UTC) and the model re-analysis ECMWF ERA-15 and ECMWF ERA-40 (at 18:00 UTC). The vertical resolution of the water vapour mixing ratio and temperature profile measurements by of the Raman lidar  are the

same as for the previous two case studies. The agreement between BASIL and the radiosonde in terms of water vapour mixing ratio is very good, with deviations not exceeding 0.5 g kg$^{-1}$ up to ~ 10 km, with the only exception of the boundary layer region where effects associated with water vapour heterogeneity are larger. Water vapour mixing ratio profiles from AIRS, IASI, ECMWF and ECMWF-ERA40 show values systematically smaller than BASIL and the radiosonde at any altitude, with deviations being large in the boundary layer (values for AIRS, IASI and ECMWF being in the range 2-6 g kg$^{-1}$,

while values of BASIL and the radiosonde being in the range 3-10 g kg$^{-1}$) and smaller (1-2 g kg$^{-1}$) above the ABL. The mean bias of BASIL vs. the radiosonde, AIRS, IASI, ECMWF and ECMWF-ERA40 are 11% (or 0.13g/kg$^{-1}$), 90% (or 0.7 g kg$^{-1}$), 80% (or 0.61 g/kg$^{-1}$), 53% (or 0.4 g/kg$^{-1}$) and 64% (or 0.49 g/kg$^{-1}$), respectively. The agreement between BASIL and the other sensors/models in terms of temperature is very good, with the mean bias of BASIL vs. the radiosonde, AIRS, IASI, ECMWF and ECMWF-ERA40 being 0.4 K, 1.17K, 1.16K, -0.58 K, and 0.51 K respectively. Both BASIL and the

radiosonde profiles reveal the presence of temperature fluctuations (well visible above 12 km) possibly associated with gravity waves propagation. The deviations between BASIL and the radiosonde profiles, on one side, and AIRS, IASI,





ECMWF and ECMWF-ERA40 profiles, on the other side, translates into large systematic deviations of the corresponding relative humidity profiles.

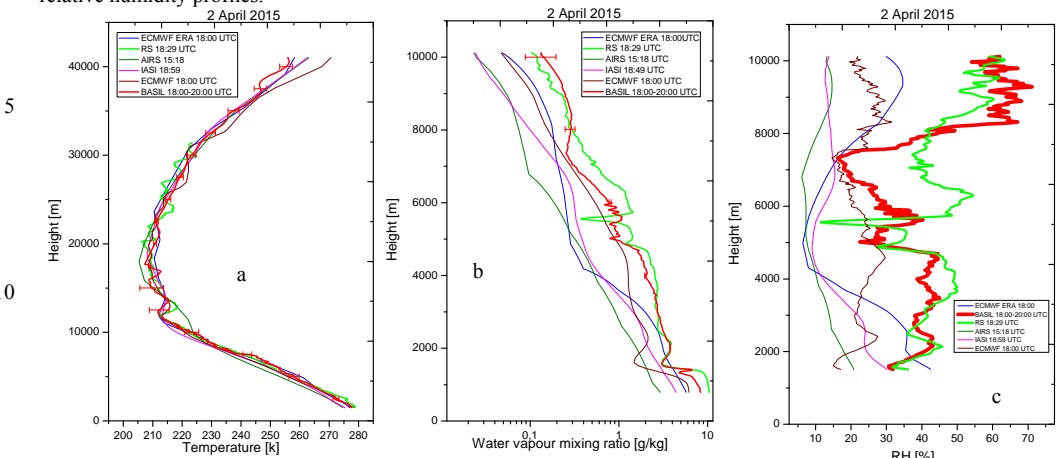

Figure 6: Mean profile of water vapour mixing ratio (a), temperature (b) and relative humidity (c) as measured by BASIL over the time period 18:00-20:00 UTC on 2 April 2015, together with the closest in time profiles from the radiosonde (at 18:29UTC), IASI, AIRS and the model re-analysis ECMWF (ERA-15 and ERA-40, all at 18:00 UTC).

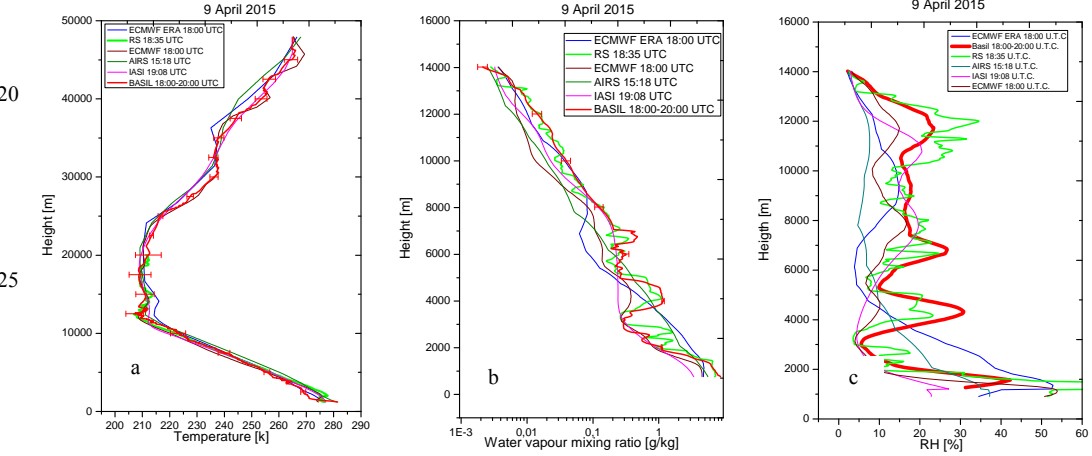

Figure 7: Mean profile of water vapour mixing ratio (a), temperature (b) and relative humidity (c) as measured by BASIL over the time period 18:00-20:00 UTC on 9 April 2015, together with the closest in time profiles from the radiosonde (at 18:35 UTC), IASI, AIRS and the model re-analysis ECMWF (ERA-15 and ERA-40, all at 18:00 UTC).




Finally, figure 7 illustrates the mean water vapour mixing ratio (a), temperature (b) and relative humidity (c) profile measured by BASIL on 9 April 2015 over the time interval 18:00-20:00 UTC, together with the simultaneous radiosonde profile (18:35 UTC) and the closest in time vertical profiles the water vapour mixing ratio profiles from IASI (at 19:08 UTC) and AIRS (at 15:18 UTC) and the model re-analysis ECMWF and ECMWF-ERA40 (at 18:00 UTC).

The agreement between BASIL and the radiosonde in terms of water vapour mixing ratio profiles is very good, with both sensors properly revealing the humidity decrease within the ABL (from 8-10 g kg$^{-1}$ down to ~1 g kg$^{-1}$) and the two elevated humidity layers between 3 and 5 km and at 7 km, the latter two possibly associated with the passage of a cold front. For this specific case study, IASI properly reproduces over a large portion of the sounded interval the water vapour mixing ratio profile observed by BASIL, but fails in correctly reproducing the large mixing ratio values observed in the ABL (IASI

values do not exceed 3.5 g kg$^{-1}$) and in capturing the two upper humidity layers. AIRS is found to overestimate BASIL, the radiosonde and IASI in the free troposphere up to ~4 km, while it underestimates these sensors above. ECMWF overestimates BASIL, the radiosonde and IASI in the free troposphere up to ~4 km, underestimates these sensors (up to 1 K) in the altitude region 4-9 km and is in good agreement with these sensors above. The agreement between BASIL and the other sensors/models in terms of temperature profiles is quite good, the comparison with the radiosonde extending only up to

~20 km because of an early blast of the balloon on this day. The water vapour mixing ratio mean bias of BASIL vs. the radiosonde, AIRS, IASI, ECMWF and ECMWF-ERA40 are -4.1% (or -0.0033 g kg$^{-1}$, 30 % (or 0.0622 g kg$^{-1}$), 38 % (or 0.09 g kg$^{-1}$), 50% (or 0.1 g kg$^{-1}$) and 40 % (or 0.09 kg$^{-1}$), respectively, while the temperature mean bias of BASIL vs. the radiosonde, AIRS, IASI, ECMWF and ECMWF-ERA40 are 0.53 K,0.72 K, 0.56 K,0.56 K and 0.7 K respectively.

**6.3 Assessment of the BIAS and RMS deviation between the different sensors/models**

The performance of the different profiling sensors and models considered in the present study are assessed through a dedicated statistical analysis. Specifically, for each sensor/model pair and each case study, the relative bias and root-mean square (RMS) deviation profiles are determined in terms of both water vapour mixing ratio and temperature. For all inter-comparisons reported in this paper, we computed bias and RMS deviation considering vertical intervals of 500 m (i.e. $z_2$-$z_1$= in expressions (16) and (17) is taken equal to 500 m). The upper level of the comparison may vary for each different

sensor/model pair and may vary from day-to-day.

Figure 8 shows the water vapour mixing ratio BIAS and RMS deviation profiles, expressed both in terms of absolute and percentage units, for all sensor/model pairs. As expected, the absolute BIAS shows larger values in the ABL (in the range -3/+5 g kg$^{-1}$), with values being typically smaller than ±1 g kg$^{-1}$ above 2 km. More specifically, in the ABL the absolute BIAS of BASIL vs. the radiosondes is in the range ±0.3 g kg$^{-1}$, with similar small values being also observed in the comparison of

BASIL vs. ECMWF-ERA40. Larger absolute BIAS values in the ABL are found to characterize the comparisons of BASIL/radiosondes/ECMWF-ERA40 vs. IASI and AIRS, with values up to 5 g kg$^{-1}$. In this regard, it is to be pointed out that, while the distance between BASIL and the radiosonde launching facility (IMAA-CNR) is only ~7 km and the grid point of ECMWF ERA-15 and ECMWF ERA-40 has a size of 9×9 km and is centered in between BASIL and IMAA-CNR,





including both sites, the distance between BASIL and IASI/AIRS footprint centers is ~25 km, these footprints having sizes of 12×12 km and 72×72 km, respectively. Consequently, when comparing BASIL/radiosondes/ECMWF-ERA40 vs. IASI and AIRS, the effects associated with water vapour heterogeneity are much more important (see figure 1).

For all sensor/model pairs, the absolute BIAS shows values smaller than ±0.1 g kg$^{-1}$ above 8 km and smaller than ±0.02 g kg$^{-1}$

above 10 km. In the altitude region 8-16 km the mutual bias of BASIL vs. the radiosondes or ECMWF is smaller than ±0.01 g kg$^{-1}$, while the mutual bias of BASIL vs. ECMWF-ERA40 and AIRS vs. IASI is smaller than ±0.07 g kg$^{-1}$ above 10 km. The mutual bias of BASIL vs. AIRS or IASI is in the range smaller than 0.01 g kg$^{-1}$ above 11 km. RMS deviation values are comparable with BIAS values for all sensor/model pairs, which testifies that statistical uncertainties and changes in the measured/modeled atmospheric parameters poorly contribute to the profiles' deviations. More specifically, for all

sensor/model pairs, the absolute RMS deviation shows values smaller than ±1.5 g kg$^{-1}$ above 2 km, smaller than ±0.1 g kg$^{-1}$ above 8 km and smaller than ±0.02 g kg$^{-1}$ above 10 km.

For all sensor/model pairs, the relative or percentage BIAS shows values in the range ±60 % all the way up to 16 km. The smallest percentage bias is found in the comparison of BASIL vs. the radiosondes, with values not exceeding ±18 % all the way up to 12 km and values in the range ±13 % above the ABL up to 4 km. A small percentage bias is also found in the

comparison of BASIL vs. ECMWF, with values not exceeding ±30 %. Positive percentage bias values in the range 0/60 % are found to characterize the comparison of BASIL/radiosondes/ECMWF-ERA40 vs. IASI and AIRS at all altitudes, which testifies that IASI and AIRS underestimate all other sensors/models. Percentage bias values in the range ±25 % are found in the comparison of IASI vs. AIRS in the altitude region 6-16 km, while larger values (up to 50 %) are found below 6 km, the agreement between the two sensors in the upper portion of the profile confirming that they both are underestimating all other

considered sensors/models (BASIL/radiosondes/ECMWF-ERA). Values of the percentage RMS deviation for the comparison of BASIL vs. the radiosondes and ECMWF are smaller than 40 % up to 11 km and smaller than 30 % above. Values of the percentage RMS deviation typically smaller than 50 %, but with sporadic values as large as 65-70 %, are found to characterize the comparison of BASIL/radiosondes/ECMWF-ERA vs. IASI and AIRS at all altitudes.

Figure 9 illustrates the temperature absolute BIAS and RMS deviation profiles for all sensor/model pairs. The BIAS of

BASIL vs. the radiosondes is in the range ±1 K above the ABL up to 12 km, with deviations in the ABL not exceeding 2 K. Except for a few points, BIAS values are within ±2 K up to 30 km, this being the maximum altitude reached by the radiosonde. The BIAS of BASIL vs. ECMWF-ERA40 is within the range ±0.8 K up to 12.5 km. For all sensor/model pairs, the BIAS shows values in the range ±5 K all the way up to 50 km. As for the water vapour mixing ratio, RMS deviation values for all sensor/model pairs are slightly exceeding BIAS values, which testifies the limited contribution of statistical

uncertainties and changes in the measured/modeled atmospheric parameters in determining the deviations between profile pairs.

The vertically-averaged mean bias, $\overline{bias}$, and RMS deviation, $\overline{RMS}$, over the entire inter-comparison range is determined through the application of the weighted mean (Bhawar $et\ al.$, 2011):



$$\overline{bias} / \overline{RMS} = \frac{\sum_{i=1}^{N} w_i (bias_i / RMS_i)}{\sum_{i=1}^{N} w_i} \qquad (19)$$

where $bias_i/RMS_i$ being the mean relative/absolute bias/RMS within the $i$th vertical interval, $w_i$ is the corresponding weight and $N$ is the number of vertical windows. $N$ may vary for the different sensor/model pairs. For the comparisons in terms of water vapour mixing ratio, which extends up to 16 km, the number of vertical windows $N$ is equal to 30. The comparisons in

terms of temperature extends up to 50 km for all different sensor/model pairs, with the number of vertical windows $N$ being equal to 98, with the only exception of the comparisons including the radiosondes, in this case the number of vertical windows $N$ being equal to 58.

The weight $w_i$ is given by the number of inter-comparisons possible within each vertical window and varies between 0 and 15, this latter value representing the total number of case studies included in this inter-comparisons effort. A weighted mean

is necessary because, in case of missing data at some specific altitude, the number of inter-comparisons may be smaller than 15 and thus data from these altitudes must have a lower weight in the vertically-averaged mean.

Table 1 includes the vertically-averaged mean absolute/percentage bias, $\overline{bias}$, and RMS deviation, $\overline{RMS}$, values for the water vapour mixing ratio inter-comparison , which includes all possible sensor/model pairs. The smallest absolute $\overline{bias}$ value is found to characterize the comparison of radiosondes vs. BASIL (0.024 g kg$^{-1}$). Small values of the absolute $\overline{bias}$ are

also found in the comparison of ECMWF and ECMWF-ERA40 vs. IASI and AIRS (ECMWF vs. IASI=-0.0295 g kg$^{-1}$, ECMWF vs. AIRS=-0.042 g kg$^{-1}$, ECMWF-ERA40 vs. IASI= 0.075 kg$^{-1}$, ECMWF-ERA40 vs. AIRS= 0,071 g kg$^{-1}$). A possible motivation behind these low values considers the fact that ECMWF reanalysis products, as those used in the present inter-comparison effort (ECMWF ERA 15 and ERA 40) heavily rely on the assimilation of IASI and AIRS data, especially in the UTLS region. This is also responsible for the small $\overline{bias}$ value of ECMWF vs. ECMWF ERA (-0.056 g kg$^{-1}$). Much

larger absolute $\overline{bias}$ values are found to characterize the comparison of BASIL/radiosondes vs. IASI, AIRS, ECMWF and ECMWF-ERA40 (BASIL vs. IASI=0.346 g kg$^{-1}$, BASIL vs. AIRS=0.342 g kg$^{-1}$, ECMWF vs. BASIL =0.297 g kg$^{-1}$, BASIL vs. ECMWF-ERA40=-0.381 g kg$^{-1}$, CNR vs. IASI=0.55 g kg$^{-1}$, CNR vs. AIRS=0.584 g kg$^{-1}$, CNR vs. ECMWF=0,3 g kg$^{-1}$, CNR vs. ECMWF-ERA40=0.318 g kg$^{-1}$). The value of absolute $\overline{RMS}$ for the comparison of radiosondes vs. BASIL is 0.148 g kg$^{-1}$, which is sensitively larger than the corresponding absolute $\overline{bias}$ values. This is most probably the effect of the large

statistical uncertainty affecting BASIL measurements in the UTLS region and the radiosonde horizontal drift, which determines humidity profile changes which are associated with different sounded air-masses. Large values of absolute $\overline{RMS}$ are also found to characterize the comparison of AIRS vs. all other sensors/models (AIRS vs. BASIL=0.536 g kg$^{-1}$, AIRS vs. CNR= 0.686 g kg$^{-1}$, AIRS vs. IASI=0.381 g kg$^{-1}$, AIRS vs. ECMWF=0.33 g kg$^{-1}$, AIRS vs. ECMWF-ERA40=0.216 g kg$^{-1}$). The motivation for these large values is the large size of AIRS footprint (72×72 km), which determines a measurement loss

of representativeness when compared to all other localized sensor/model data. Large values of absolute $\overline{RMS}$ are also found





to characterize the comparison of IASI vs. all other sensors/models (IASI vs. BASIL=0.403 g kg$^{-1}$, IASI vs. CNR= 0.63 g kg$^{-1}$, IASI vs. ECMWF=0.165 g kg$^{-1}$, IASI vs. ECMWF-ERA40=0.252 g kg$^{-1}$). Such large values are possibly associated with the considerable distance between IASI footprint and all other sensors and models, especially in the presence of horizontal heterogeneities in the humidity field. The large absolute $\overline{RMS}$ values characterizing the comparisons of BASIL/radiosondes

vs. ECMWF/ECMWF-ERA40 are again possibly associated with the limited effectiveness of these re-analyses within the ABL, where most humidity is located, as well as with their poor effectiveness in the UTLS region.

Values of the percentage $\overline{bias}$ confirm most of the considerations above. It is to be pointed out that percentage $\overline{bias}$ is quantity very sensitive to the variability of the data in the UTLS region, more than the absolute $\overline{bias}$, as in fact the water vapour mixing ratio has a large variability within the troposphere, varying over four orders of magnitude from the surface to

the UTLS region. A very small percentage $\overline{bias}$ value is found to characterize the comparison of radiosondes vs. BASIL (3.85%), this value being in good agreement with the absolute $\overline{bias}$ value and testifying the accuracy and agreement of these two sensors throughout the sounded vertical interval, especially in the ABL and in the UTLS. Small values of the percentage $\overline{bias}$ are also found to characterize the comparison of ECMWF/ECMWF-ERA40 vs. IASI/AIRS (ECMWF vs. IASI= -2.25 %, ECMWF vs. AIRS=-2.19 %, ECMWF-ERA40 vs. IASI= 18 % ECMWF-ERA40 vs. AIRS= 20 %). Again, a possible

motivation behind these low values is the fact that ECMWF reanalysis products (ECMWF ERA 15 and ERA 40) strongly rely on IASI and AIRS data, especially in the UTLS region. Much larger values of the percentage $\overline{bias}$ are found to characterize the comparison of BASIL/radiosondes vs. IASI, AIRS, ECMWF and ECMWF-ERA40 (BASIL vs. IASI=37.52 %, BASIL vs. AIRS=36.84 %, BASIL vs. ECMWF=-25 % BASIL vs. ECMWF-ERA40= 31 %, CNR vs. IASI=56 %, CNR vs. AIRS=64 %, CNR vs. ECMWF=55 %, CNR vs. ECMWF-ERA40=36 %). Coming to the values of the percentage $\overline{RMS}$,

for the comparison of BASIL vs. the radiosondes this is 23 %, which is larger than the absolute $\overline{bias}$, again probably because of the large statistical uncertainty affecting BASIL measurements in the UTLS region and the radiosonde horizontal drift. Large value of the percentage $\overline{RMS}$ also characterize the comparison of AIRS and IASI with all other sensors/models (AIRS vs. BASIL= 59.5 %, AIRS vs. CNR= 64 %, AIRS vs. IASI=37.3 %, AIRS vs. ECMWF= 40,7, AIRS vs. ECMWF-ERA40= 40.7 %, IASI vs. BASIL= 44.1 %, IASI vs. CNR= 58.6 %, IASI vs. AIRS= 37.2 %, IASI vs. ECMWF= 31.7 %, IASI vs.

ECMWF-ERA40= 37.3 %), possible motivations for which having already been illustrated above when discussing the absolute $\overline{RMS}$.

Table 2 includes the vertically-averaged mean absolute bias and RMS deviation values for the temperature inter-comparison for all considered sensor/model pairs. It is to be specified that, while an estimate of the percentage bias and RMS deviation is necessary for the adequate assessment of quality (accuracy/precision) of the water vapour mixing ratio

measurements/analyses, as in fact this quantity may vary more than four orders of magnitude in the altitude interval considered in this inter-comparison effort (0-16 km), there is no need for an estimate of the percentage bias and RMS deviation characterizing temperature measurements/analyses, which are characterized by a much smaller variability (not





exceeding 30 %) in the considered interval (0-50 km), with the largest values at surface (typically 280-300 K) and the smallest values at the tropopause (typically 200-210 K). The smallest absolute $\overline{bias}$ values characterize the comparison of radiosondes vs. BASIL (0.04 K) and ECMWF-ERA40 vs. IASI (-0.087 K). Small $\overline{bias}$ values are also found in the comparison of BASIL vs. ECMWF (-0.14 K), BASIL vs. ECMWF-ERA40 (0.62 K), BASIL vs. IASI (0.48 K), CNR vs.

ECMWF (0.57 K), CNR vs. ECMWF-ERA40 (0.33 K) and CNR vs. IASI (0.75 K). Additional small $\overline{bias}$ values are also found to characterize the comparison of ECMWF vs. ECMWF-ERA40 (0.58 K) and ECMWF vs. IASI (0.64 K). Larger $\overline{bias}$ values are found to characterize the comparison of AIRS with all other sensors/models (BASIL vs. AIRS = 1.99 K, CNR vs. AIRS =-1.18, AIRS vs. IASI=-1.49 K, ECMWF vs. AIRS=2,14 K and ECMWF-ERA40 vs. AIRS =1,19 K), with AIRS always underestimating all other sensors and models. The above results reveal, with the only exception of AIRS, a

very good agreement between all sensors and a remarkable capability of the considered models to reproduce the measured temperature profiles. As clearly shown by the bias profiles in figure 9a, most part of the bias between AIRS and all other sensors/models is found above 37 km, which reveal a negative systematic uncertainty affecting AIRS temperature profile measurements above this altitude (AIRS underestimating all other sensors/models) up to 5 K. Part of this bias is also to be attributed to the fact that AIRS slightly underestimates all other sensors/models around the tropopause. Additionally, the

small bias characterizing the comparisons of IASI with all other sensors/models testify the very good performance of this sensor in terms of temperature profile measurements and its correct assimilation in ECMWF analyses. A low value of the $\overline{RMS}$ deviation characterize the comparison of BASIL vs. the radiosondes (1.61 K). This low value is to be attributed to the fact that the comparison between BASIL and the radiosondes only extends up to 30 km and consequently the effects associated with the large statistical fluctuations affecting BASIL signals in the 30-50 km region, with the sounding of

different air-masses and with gravity waves propagation are sensitively reduced. Small $\overline{RMS}$ values are also found in the comparison of BASIL vs. ECMWF (2.06 K), BASIL vs. ECMWF-ERA40 (2.4 K), ECMWF vs. ECMWF-ERA40 (1.35 K), ECMWF vs. IASI (1.88 K) and ECMWF vs. ECMWF-ERA40 (1.64 K). Again, all comparisons involving AIRS are characterized by large values of $\overline{RMS}$, again to be largely attributed to the systematic uncertainty affecting AIRS above 37 km and leading to an underestimation of all other sensors/models.

**6.4 Overall bias affecting all sensors/models**

Making use of the available statistics of comparison results, an approach is considered to determine the overall bias values for all sensors/models involved in this inter-comparison effort. This approach, originally proposed by Behrendt *et al.* (2007a,b), can be applied in case there is at least one sensor whose measurements are comparable with all other sensors/models. For the purpose we consider the Raman lidar BASIL. Assuming equal weight on the data reliability of each

sensor/model, an estimate of the overall bias affecting all sensors/models is obtained by imposing that the summation of all mutual biases between sensor/model pairs is equal to zero. The choice of attributing equal weight to the data reliability of each sensor/model is driven by the awareness that none of them can *a priori* be assumed more accurate than the others.



Based on this approach, the overall absolute bias affecting water vapour profile data from BASIL, the radiosondes, IASI, AIRS, ECMWF and ECMWF-ERA40 is estimated to be 0.004 g kg$^{-1}$, 0.021 g kg$^{-1}$, -0.350 g kg$^{-1}$, -0.346 g kg$^{-1}$, 0.293 g kg$^{-1}$ and 0.377 g kg$^{-1}$, respectively, as sketched in figure 10a. Very similar values are obtained by assuming that the radiosonde, which was the sensor used for the calibration of the Raman lidar and is traditionally used for validation/calibration of

humidity profiles from other sensors or models, is bias-free. In this case the overall absolute bias affecting the water vapour mixing ratio profile from BASIL, IASI, AIRS, ECMWF and ECMWF-ERA40 is 0.008 g kg$^{-1}$, -0.345 g kg$^{-1}$, -0.341 g kg$^{-1}$, 0.0298 g kg$^{-1}$ and 0.382 g kg$^{-1}$, respectively. Both approaches lead to an overall bias affecting water vapour profile measurements from BASIL and the radiosondes smaller than 0.02 g kg$^{-1}$.

The same approach was applied to determine the overall absolute temperature bias for all sensors/models involved in this

inter-comparison effort. In this case, as we previously identified a significant systematic uncertainty affecting AIRS measurements, this sensor was excluded from the summation of all mutual biases between sensor/model pairs. Thus, assuming equal weight on the data reliability of all other sensor/model the overall absolute bias affecting temperature profile data from BASIL, the radiosondes, IASI, AIRS, ECMWF and ECMWF-ERA40 is found to be 0.30 K, -0.34 K, 0.18 K, -1.63 K, -0.16 K and 0.32 K, respectively, as sketched in figure 10b. These results confirm   that AIRS systematically

underestimates all other sensors and models. Very similar values are obtained by assuming that the radiosonde is bias-free. In this case the overall absolute bias affecting the temperature profile from BASIL, IASI, AIRS, ECMWF and ECMWF-ERA40 is 0.14 K, 0.02 K, -1.95 K, -0.32 K and 0.16 K, respectively. Both approaches lead to an overall bias affecting temperature profile measurements from BASIL and the radiosondes smaller than ± 0.35 K.



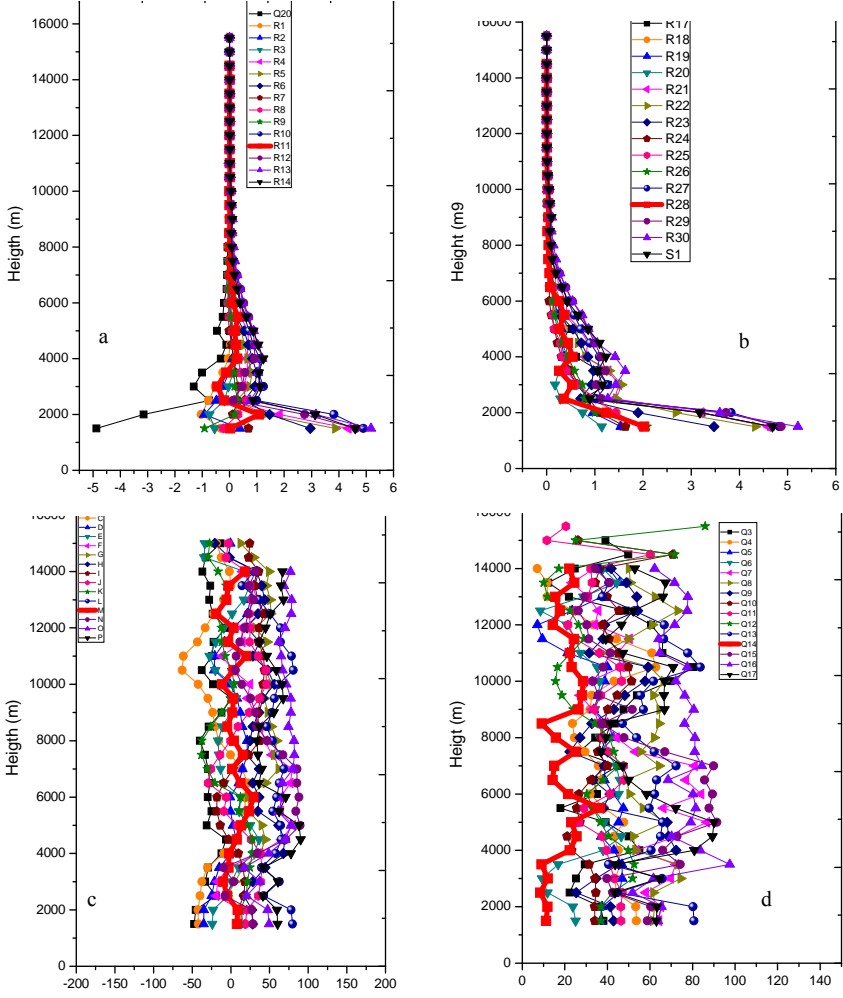

Figure 8: Vertical profiles of water vapour mixing ratio mean BIAS and RMS deviation for all sensor/model pairs: (a)

5    absolute BIAS, absolute RMS (b) percentage BIAS (c) and percentage RMS (d).



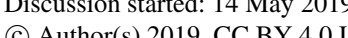

Figure 9: Vertical profiles of temperature mean BIAS and RMS deviation for all sensor/model pairs: (a) absolute BIAS, absolute RMS (b).

| | CNR vs. ECMWF ERA | ECMWF ERA vs. AIRS | ECMWF ERA vs. IASI | CNR vs. AIRS | RS vs IASI | AIRS vs IASI | ECMWF vs ECMWF ERA | ECMWF vsAIRS | ECMWF vs IASI | BASIL vsECMWF ERA | RS vs BASIL | BASIL vs AIRS | BASIL vs IASI | ECMWF vs BASIL | RS vsECMWF |
|---|---|---|---|---|---|---|---|---|---|---|---|---|---|---|---|
| Abs.BIAS (g kg⁻¹) | 0,318 | 0,071 | 0,075 | 0,584 | 0,55 | 0,006 | -0,056 | -0,042 | -0,029 | -0,381 | 0,0245 | 0,342 | 0,346 | 0,297 | 0,3 |
| Abs. RMS (g kg⁻¹) | 0,466 | 0,216 | 0,252 | 0,686 | 0,63 | 0,381 | 0,159 | 0,33 | 0,165 | 0,576 | 0,148 | 0,536 | 0,403 | 0,324 | 0,343 |
| Perc. BIAS (%) | 36,903 | 20,483 | 18,686 | 64,562 | 56,33 | 1,591 | -16,562 | -2,19 | -2,257 | -31,091 | 3,857 | 36,84 | 37,521 | 25,034 | 55,431 |
| Perc. RMS (%) | 53,768 | 37,066 | 37,343 | 73,933 | 58,696 | 37,275 | 35,6 | 40,761 | 31,776 | 52,982 | 19,511 | 59,455 | 44,101 | 30,93 | 58,228 |

Table 1: Water vapour mixing ratio vertically-averaged mean absolute/percentage bias and RMS deviation values for all considered sensor/model pairs.

| | ECMWF ERA vs. CNR | ECMWF ERA vs. AIRS | ECMWF ERA vs. IASI | CNR vs. AIRS | RS vs IASI | AIRS vs IASI | ECMWF vs ECMWF ERA | ECMWF vsAIRS | ECMWF vs IASI | BASIL vsECMWF ERA | RS vs BASIL | BASIL vs AIRS | BASIL vs IASI | ECMWF vs BASIL | RS vsECMWF |
|---|---|---|---|---|---|---|---|---|---|---|---|---|---|---|---|
| BIAS (K) | 0,33 | 1,19 | -0,087 | 1,18 | 0,75 | -1,49 | 0,58 | 2,14 | 0,64 | 0,62 | 0,04 | 1,99 | 0,48 | -0,14 | 0,57 |
| RMS (K) | 1,85 | 2,81 | 1,64 | 2,99 | 2,98 | 3,08 | 1,35 | 3,8 | 1,88 | 2,4 | 1,61 | 4,8 | 2,26 | 2,06 | 1,62 |

Table 2: Temperature vertically-averaged mean absolute/percentage bias and RMS deviation values for all considered sensor/model pairs.





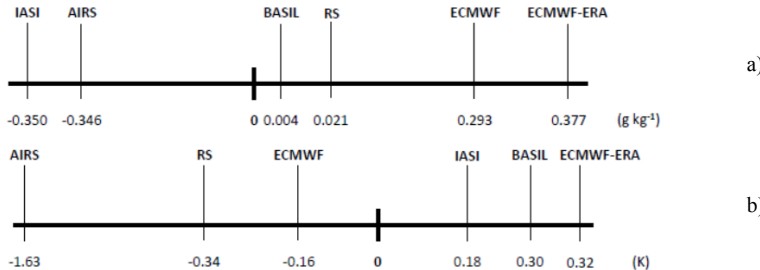

Figure 10: Overall absolute bias affecting water vapour profile (a) and temperature (b) data.

### 6.5 Assessment of the BIAS and RMS deviation: specific dedicated inter-comparison effort between BASIL and the radiosondes

Besides the results already illustrated in the earlier part of this section, a specific inter-comparison between BASIL and the radiosondes launched from IMAA-CNR (7 km away) was carried out in the period 9 October 2014- 7 May 2015, including all coincident measurements. An overall number of 11 comparisons were possible. Routine radiosonde launches started at IMAA-CNR only on October 2014, so inter-comparisons before then were very rare. Figure 11 illustrates the vertical profiles of water vapour mixing ratio and temperature mean BIAS and RMS deviation for the 11 considered comparisons. For what concerns the water vapour mixing ratio measurements, above the planetary boundary layer and up to 8.5 km (figure 11a), the mean BIAS is not exceeding ± 0,25 g/kg (or ± 10 %). Even at high altitudes bias values are very low as in fact above 8.5 km this is not exceeding ± 0,06 g/kg (or ± 50 %). For what concerns the temperature measurements, above the planetary boundary layer and up to 9.5 km, absolute biases are within ± 1 K.



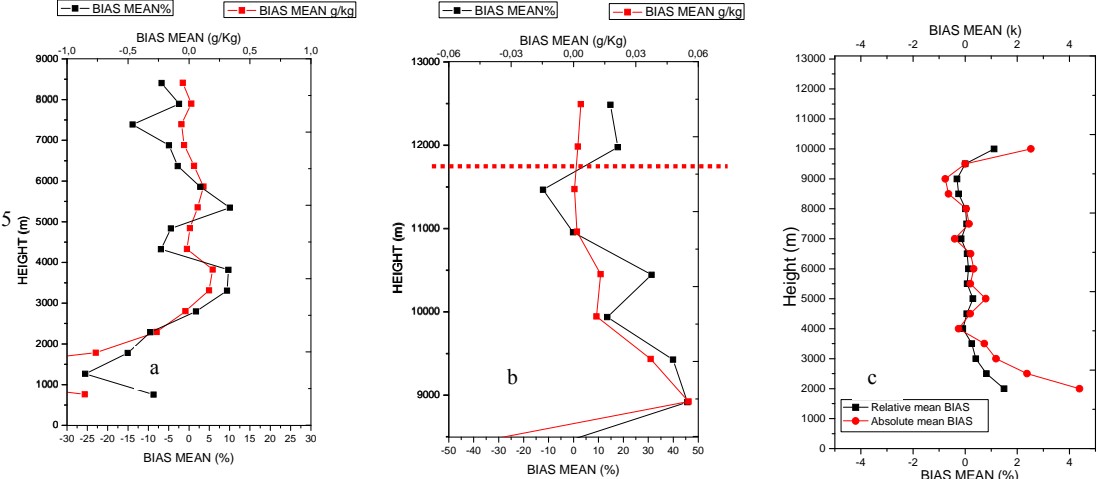

Fig. 11: Vertical profiles of water vapour mixing ratio and temperature mean BIAS and RMS deviation for the 11 comparisons of BASIL vs. the radiosondes available in the the period 9 October 2014- 7 May 2015: (a) water vapour mixing ratio: vertical interval 0-8.5 km; (b) water vapour mixing ratio: vertical interval 8.5-13 km; c) temperature: vertical interval 0-13 km. The dashed red line in panel b represents the mean tropopause altitude.

**7 Summary**

Case studies illustrated in this paper demonstrate the ability of BASIL to perform temperature profile measurements up to 50 km and water vapour mixing ratio profile measurements up to 15 km, considering an integration time of 2 hours and a vertical resolution of 150 m, with a systematic measurement uncertainty (bias) not exceeding 0.1 K and 0.1 g kg$^{-1}$, respectively. Temperature and water vapour profile measurements carried out by BASIL are compared with profiles from a variety of other sensors/models, namely radiosondings, the satellite instruments IASI and AIRS and with model re-analyses data (ECMWF and ECMWF-ERA). Comparisons between BASIL and the different sensor/model data in terms of water vapour mixing ratio indicate a mean absolute/relative mutual bias of -0.024 g kg$^{-1}$(or -3.9 %), 0.346 g kg$^{-1}$ (or 37.5 %), 0.342 g kg$^{-1}$(or 36.8 %), -0.297 g kg$^{-1}$ (or -25 %), -0.381 g kg$^{-1}$ (or -31 %), when compared with radisondings, IASI, AIRS, ECMWF, ECMWF-ERA, respectively. Comparisons in terms of temperature measurements reveal a mean absolute mutual bias between BASIL and the radisondings, IASI, AIRS, ECMWF, ECMWF-ERA of -0.04, 0.48, 1.99, 0.14, 0.62 K, respectively. Larger temperature biased are found between AIRS and all other sensors/models, which is the result of AIRS slightly underestimating all other sensors/models around the tropopause and above 37 km.





The possibility to assess the overall bias values for all sensors/models included in this inter-comparison effort was also exploited, benefiting from the circumstance that the Raman lidar BASIL could be compared with all other sensor/model data. The overall absolute bias affecting water vapour/temperature profile data from BASIL, the radiosondes, IASI, AIRS, ECMWF and ECMWF-ERA40 was estimated to be 0.004 g kg$^{-1}$/0.30 K, 0.021 g kg$^{-1}$/-0.34 K, -0.35 g kg$^{-1}$/0.18 K, -0.346 g

5  kg$^{-1}$/-1.63 K, 0.293 g kg$^{-1}$/-0.16 K and 0.377 g kg$^{-1}$/0.32 K.

The present study allows us to get confidence on the high quality of the water vapour and temperature profiling carried out by BASIL and included in the NDACC database and on the possibility to use long-term records of these measurements for monitoring of atmospheric composition and thermal structure changes and, ultimately, for climate trend studies.

**Author contribution**

10  Paolo Di Girolamo designed the experiment and together with Benedetto De Rosa and Donato Summa carried out the measurements. Benedetto De Rosa and Donato Summa developed the data analysis algorithms and the former of the two carried out the data analysis. Benedetto De Rosa and Paolo Di Girolamo prepared the manuscript with contributions from Donato Summa.

**Acknowledgements**

15  This work was possible based on the support from the Italian Ministry for Education, University and Research under the Grant OT4CLIMA.We thank GCOS Reference Upper-Air Network (GRUAN) for the provision of the  radiosonde data. I specific thank is given to IMAA-CNR of Tito Scalo and the site representative Dr. Fabio Madonna.



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
