# Peer review of "Temperature and water vapour measurements in the framework of NDACC"

_Atmospheric Measurement Techniques, 2019_

## Referee Comment (RC1) · Anonymous Referee #1 · 25 Jul 2019

This article presents the uncertainties associated with the BASIL lidar following its entry into the NDACC network. The lidar operated on a weekly basis between November 7, 2013 and October 5, 2015. The announced accuracies for the lidar system are extremely high with biases of 0.1 K and 0.1 g kg-1 for measurements between the surface and 15 km altitude for water vapour, and between the surface and 50 km altitude for temperature. These values are obtained for a temporal resolution of 2 hours and a vertical resolution of 150 m, both day and night.

While the BASIL instrument is of undeniable interest to the international scientific community, it is not presented here in a relevant way.

I find that biases are underestimated and the approach to estimating them needs to be clarified. The main points that make me doubt the results are listed in the following:

[Figure]

- The profiles used to calibrate the lidar are not explained, how many times this calibration had to be repeated during the measurement period. How stable is the calibration over time? Is there not an influence of the aging of the components, of the effects of temperature, for example during the succession of seasons?

- If the lidar has been calibrated compared with modelling data, is it not normal that the biases are small? AIRS, IASI, radiosonde and model data are not independent. AIRS and IASI operate on a similar way with average kernel leading to ~2x higher vertical resolution for IASI. Radiance data are assimilated into the model, just like radiosonde data, so the reference profiles are not independent.

- How can such low biases be explained given that the profiles referred to are associated with much higher biases. For example, statistical studies on a large number of radiosondes have shown bias in the order of 0.4 g kg-1 and 0.5 K. For IASI we are on 0.5-1 g kg-1 and 0.5-1 K depending of the kernel averaging function.

- Why limit yourself to 4 case studies? This considerably limits the investigation and makes the statistical study unrepresentative. The argument developed at the beginning of section 6.1 is not relevant.

- The BASIL lidar has integrated the NDACC network which already contains other water vapour and temperature lidars. It would therefore relevant to compare the accuracy of these different lidars with that of BASIL.

- To improve the clarity of the study, 2 sets of profiles should be defined. The first set would be dedicated to calibration and the second to the study of biases and standard deviations.

The authors place their work so heavily that they forget the existence of works conducted by other teams around the world. There were other NDACC publications, other cross-comparison exercises using Raman technology, very extensive studies on the representativeness of radiosondings and comparisons to modelling.
Specific comments

Abstract. It must be revised after taking into account the previous remarks. Repetitions should be avoided (L17). The end of the abstract is reminiscent of circular reasoning. Can an absolute bias be negative?

Introduction P2L21. The cost of a radiosonding is about 250 €Ìs a lidar competitive at this level, especially for the upper troposphere and stratosphere? P3L5. Perhaps there is an overlap factor? P3L10-11. Not necessary because it is not a paper topic.

Section 2 It would be interesting to have a table that summarizes the main characteristics of the lidar. P3L28 and 29. Typography

Section 3.1 P5L5. There are much earlier references. P5L12. For numerical weather forecasting, IASI inversions are not used, radiances are assimilated directly.

Sections 4.1, à 4.3 These sub-sections are already very well known, it is enough to highlight the sources in order to simplify the article. The important point is the calibration which needs to be clarified. The choice of a two-parameter temperature adjustment function must be justified. This type of adjustment does not guarantee optimal accuracy, it is preferable to use functions with 3 parameters, especially with a wide temperature range. It is surprising to obtain such low biases with this type of function.

The method developed by Auchecorne et al. is already well described and the error sources have been seriously studied and evaluated. Instead, errors should be discussed in this section because the results are very dependent on the lidar used (e.g. the optical filtering technique used). The methods developed by the pioneering authors are to be considered but applied to the BASIL lidar. There is no reason to achieve the same levels of error.

Section 4.2.2 P9L20. How is this altitude interval justified? This is a critical point to reduce the uncertainty random and it must be justified for each lidar. P10L6. Maybe a little less, 80 km on average? P10L10. It is not the good term P10L23. It is mainly

due to the stability of the equation which limits the error propagation. Equation 16. Subscripts are missing Equation 17. Idem. Are you sure about this relative RMS calculation? You do a simple average on RMS. P12L12. What type of interpolation? P12L13. The bias is signed, so it is not absolute. P13L14-15. Already explained above. The first paragraph of section 6.1 should be put before. P14L6. How are such values calculated? P14L21. What type of numerical filter? P21L9. There have already been cross-comparison studies of IASI and Raman lidar that reveal such differences in the ABL; differences due to the characteristics of IASI. Maybe you should talk about it.

Section 6.3. This paragraph is cumbersome to read and could be greatly reduced with a better synthesis.

Acknowledgments Many acknowledgments are missing for the data sets used, whether spaceborne missions or modelling.

---

## Referee Comment (RC2) · Anonymous Referee #2 · 8 Aug 2019

The paper presents a thorough assessment of the performance of the Raman lidar BASIL operated in Potenza, through a series of intercomparison with a variety of sensors (in situ and passive remote) and numerical weather prediction models. This commendable effort is conducted on the BASIL dataset acquired since November 2013, i.e. after BASIL has integrated the NDACC network. The paper focuses on comparisons with q et T profiles from radiosoundings launched from a nearby met station, as well as q and T profiles derived from the IASI and AIRS sensors in the vicinity of Potenza. Comparison with q and T profiles from 2 analyses from the ECMWF NWP model are also included. The results contained in this paper are of interest to the NDACC community. Nevertheless, the paper is tedious to read, which may in part be related to the fact that the authors present several intercomparison periods (4 case studies and

2 longer periods) for which biases, deviations are discussed at length in both absolute and relative values. My understanding is that the comparison with the radiosounding data from the station nearby Potenza is the key for a proper calibration of the Raman system. I think that the authors should start presenting this aspect thoroughly first before declining the comparison in the framework of 4 case studies... However, I am under the impression that radiosoundings may not be the only datasets used to "calibrate" the Raman retrievals (e.g. the work conducted for the case study on 7 November 2013 for which the authors state that there were no radiosounding data available). I would encourage the authors to clarify this in the revised manuscript. Are they using reanalyses products for calibration of BASIL? In spite of the interest of such paper, the paper should be improved with respect to the points below: - Why is the intercomparison limited to the first 2 years of the participation of BASIL to NDACC? - In the abstract and in the summary, it should be mentioned that the bias values for the entire T and q profiles. Also, how do you reconcile the numbers at lines 20-21, 23 and 26 with those at line 12? - Regarding the vertical resolution of the q profiles: in the abstract and summary it is just mentioned 150 m, whereas in the text in Section 6 (Case study 7 Nov 2013, p14) the resolution is stated to be 300 m between 6 and 8 km and 600 m above 8 km. The Same holds for the vertical resolution of the T profiles: in the abstract and summary it is just mentioned 150 m, whereas in the text in Section 6 (p15) the resolution is stated to be 600 m above 6 km. - What is the interest of comparing BASIL products with IASI and AIRS products, especially since they are assimilated in NWP model reanalyses products? - P7: lines 4-5: assessment of K(z) up to 15 km is crucial here to derive the performance of BASIL. You need to say more. How many soundings were used? How do you manage to assess a K(z) up to 15 km with a met sonde that is drifting away from the launch point because of wind? What kind of humidity sound were used for the RDS? Most (if not all) of the commercial sondes are known to have issues with measurement in low humidity conditions.... - P8, lines 13-14: a and b are determined from co-located soundings? How do you deal with a met sonde that is drifting away from the launch point because of wind? Up to what altitude 25 km.. how do

you ensure a and b are not offset by the soundes drifting? Also what is the sensitivity of the T(z) retrievals on a and b retrievals? - P9, line 4: The integration technique is designed to retrieve T profiles above 20 km... why do you say below 20 km here? - 7 Nov 2013 Case: What do you use to assess the BASIL calibration if there is no RDS? Line 21: what kind of smoothing filter? How do you achieve 150 m when the resolution of the 2-h profile is 300 or even 600 m above 6/8 km? - P21: Section 6.3, line 23: now the vertical resolution of the profiles is 500 m... not 150 m? line 27: what are all sensor/model pairs? how many pairs for each type of comparisons? What period does this cover? - Section 6.5 p 29: Why only the period 9 October 2014- 7 May 2015? Are 11 comparisons enough? Why not do this for the entire period starting with BASIL entering the NDACC network?
* * *

---

## Author Comment (AC2)

Dear Editor,

We are very grateful to the two referees for their appropriate and constructive suggestions and for their proposed corrections. We have addressed all issues raised and have modified the paper accordingly. If you and the referee agree on that, we are also ready to submit a revised version of the paper where all these changes have been incorporated. We believe that, thanks to their precious inputs, the quality of the manuscript has now sensitively improved. Below is a summary of the changes we made and our specific responses to the referees' comments and recommendations.

**Summary of the changes**
**(in black is the original comments of the referee and in red our responses)**

**Anonymous Referee #1**

This article presents the uncertainties associated with the BASIL lidar following its entry into the NDACC network. The lidar operated on a weekly basis between November 7, 2013 and October 5, 2015. The announced accuracies for the lidar system are extremely high with biases of 0.1 K and 0.1 g kg-1 for measurements between the surface and 15 km altitude for water vapour, and between the surface and 50 km altitude for temperature. These values are obtained for a temporal resolution of 2 hours and a vertical resolution of 150 m, both day and night.
While the BASIL instrument is of undeniable interest to the international scientific community, it is not presented here in a relevant way. I find that biases are underestimated and the approach to estimating them needs to be clarified.

We agree with the reviewer concerning the non-effective and relevant way BASIL and its performance are illustrated in the manuscript. Reported results need to be better clarified. Indeed, the statement in the Abstract which refers to an accuracy, expressed in terms of bias, of 0.1 K for temperature between the surface and 50 km and of 1 gkg$^{-1}$ for water vapour mixing ratio between the surface and 15 km may appear misleading. This information had been introduced in an incorrect way, without properly justifying and commenting it. Here we refer to the "measurement accuracy" and not to the "measurement bias" not exceeding 0.1 K and 0.1 g kg$^{-1}$", as now properly specified in the text. Here we use the term "accuracy" to refer to the combined effect on measurement performance of both the random and the systematic error. As properly illustrated in the points below, the Raman lidar is calibrated against the radiosondes, considering an altitude region (2.5-4 km), where both sensors (Raman lidar and radiosondes) have higher performances and sound the same air-masses. The measurement accuracy not exceeding 0.1 K and 0.1 g kg$^{-1}$, which we mention in the Abstract, is computed from the mutual bias and RMS deviation between BASIL and the radiosondes, used as reference for its calibration, in the upper troposphere region, whose mean value is not exceeding 0.1 g kg$^{-1}$ for water vapour mixing ratio measurements and 0.1 K for temperature measurements. This aspect has been now better clarified in the text, where the corresponding sentence has been changed as follows: "Measurements illustrated in this manuscript demonstrate the ability of BASIL to perform measurements of the temperature profile up to 50 km and of the water vapour mixing ratio profile up to 15 km, when considering an integration time of 2 h and a vertical resolution of 150-600 m, with measurement mean accuracy, determined based on comparisons with simultaneous and co-located radiosondes, of 0.1 K and 0.1 g kg$^{-1}$, respectively, up to the upper troposphere."

Furthermore, the low bias values reported in the second paragraph of the abstract are for the vertically-averaged mean bias, which indeed has a value smaller than the single bias values at different altitudes. This is because, in the vertically-averaged bias, positive and negative values present at the different altitudes average out. This approach, while debatable, has been used in a variety of previous papers, among others by Whiteman et al. 2006, Behrendt et al., 2007a,b, Bhawar

et al. 2011. Indeed the bias is an altitude dependent quantity, with values tending to be higher at higher altitudes. This is especially true when considering the mutual biases between the radiosondings and any of the sensors/models as a results of the horizontal drift of the radiosondes, which increases at increasing altitudes. In the revised version of the paper, in addition to the vertically-averaged mean bias, $\overline{bias}$, i.e. expression (17) in the paper, originally used by Behrendt et al. (2007a) and Bhawar *et al.* (2011), we decided to also introduce the vertically-averaged absolute bias, defined as the weighted mean of the absolute values in mathematical sense, or moduli, of the single bias values at different altitudes, i.e.:

$$\overline{|bias|} = \frac{\sum_{i=1}^{N} w_i |bias_i|}{\sum_{i=1}^{N} w_i}$$

The vertically-averaged absolute bias $\overline{|bias|}$ may probably appear as a more appropriate parameter to quantify the vertically-averaged absolute bias, as values with different signs would not cancel out. Values of $\overline{|bias|}$, now reported throughout the manuscript, are larger than the corresponding $\overline{bias}$ values.

These aspects are now better specified in the text, where the following new sentences have been introduced: "So far we have reported and discussed the mutual bias and RMS deviation profiles between different sensors/models, highlighting the altitude variability of these quantities. However, in order to assess sensors and models performance is often preferable to use a single bias/RMS deviation value. This leads us to the definition of the vertically-averaged mean bias and the vertically-averaged absolute mean bias."

Further down in the text, we also introduced the following new sentences: "The vertically-averaged absolute mean bias, $\overline{|bias|}$, and RMS deviation, $\overline{|RMS|}$, defined as the weighted mean of the moduli of the single bias values at different altitudes:

$$\overline{|bias/RMS|} = \frac{\sum_{i=1}^{N} w_i |bias_i/RMS_i|}{\sum_{i=1}^{N} w_i}$$

(18)

In the vertically-averaged absolute mean bias $\overline{|bias|}$, values at different altitudes with different signs will not cancel out. Consequently, values of $\overline{|bias|}$ are larger than the corresponding $\overline{bias}$ values."

In response to the above Reviewer's comment, we also need to specify that the claimed measurement performance is not achieved both day and night, but only refers to night-time measurements, as in fact all measurements reported and discussed in this paper for the purpose of assessing Raman lidar measurement performance are carried out at night.

The main points that make me doubt the results are listed in the following:

- The profiles used to calibrate the lidar are not explained, how many times this calibration had to be repeated during the measurement period. How stable is the calibration over time? Is there not an influence of the aging of the components, of the effects of temperature, for example during the succession of seasons?

The Raman lidar has been calibrated based on an extensive comparison with the radiosondes launched from the nearby station of IMAA-CNR. Launched radiosondes are manufactured by Vaisala (model: RS92-SGP). It is to be specified that this is the most appropriate approach we could consider, as in fact the radiosonde launching station is only 8.2 km away from the Raman lidar station and this limited distance gives confidence of the possibility to compare Raman lidar and radiosonde humidity profiles above the boundary layer. In fact, in clear sky conditions, the horizontal homogeneity of the humidity field above the boundary layer top is sufficiently high to allow assuming that the two systems (the Raman lidar and the radiosonde) are sounding the same

air masses. More specifically, for the purpose of determining the calibration coefficient, the Raman lidar and radiosonde profiles are compared over the altitude interval 2.5-4 km. Within this altitude interval, while we assume water vapour heterogeneity to be small, we also have strong Raman lidar signals and consequently high signal-to-noise ratios and small statistical uncertainties. At the same time, within this low level altitude interval, the radiosonde horizontal drift from the vertical of lidar station is limited. This reduces the chances that the two sensors are sounding different air masses. The calibration coefficient is obtained through a best-fit procedure applied to the Raman lidar and radiosonde data, the value of the coefficient being determined by minimizing the root mean square deviation between the single data points from the two profiles within the altitude interval 2.5-4 km. As the Raman lidar and the radiosonde data have different altitude arrays, for the purpose of applying the best-fit algorithm, radiosonde data are interpolated to the Raman lidar altitude levels.

These aspects are now clearly specified in the text, where the following sentences have been introduced: "The Raman lidar has been calibrated based on an extensive comparison with the radiosondes launched from the nearby station of IMAA-CNR, which is only 8.2 km away from the Raman lidar. Launched radiosondes are manufactured by Vaisala (model: RS92-SGP). For the purpose of determining the calibration coefficient the Raman lidar and radiosonde profiles are compared over the altitude interval 2.5-4 km, i.e. above the boundary layer. In fact, in clear sky conditions, the horizontal homogeneity of the humidity field above the boundary layer top is sufficiently high to allow assuming that the Raman lidar and the radiosonde are sounding the same air masses. Within this altitude interval, Raman lidar signals are strong and characterized by high signal-to-noise ratios and small statistical uncertainties. At the same time, within this low level altitude interval, the horizontal drift of the radiosonde with respect to the vertical of lidar station is limited, so that again the two sensors can be actually assumed to be sounding the same air masses. The calibration coefficient is obtained through a best-fit procedure applied to the Raman lidar and radiosonde data, the value of the coefficient being determined by minimizing the root mean square deviation between the single data points from the two profiles within the altitude interval 2.5-4 km. As the Raman lidar and the radiosonde data have different altitude arrays, for the purpose of applying the best-fit algorithm, radiosonde data have been interpolated to the Raman lidar altitude levels."

Unfortunately radiosonde launches are available in Tito Scalo only once per week, on Thursday evening. An extensive inter-comparison effort has been performed with the purpose of calibrating the Raman lidar. A mean value of the calibration constant has been determined by averaging the single calibration coefficient values from all inter-comparisons. The uncertainty affecting the calibration coefficient has been estimated as the standard deviation all single calibration values from the mean value. These aspects are now clearly specified in the text, where the following sentences have been introduced in the new section 6.1.(Raman lidar calibration): "For the purpose of determining the calibration constant $c$, a specific inter-comparison effort between BASIL and the radiosondes launched from IMAA-CNR was carried out in the period 9 October 2014-7 May 2015. An overall number of 11 comparisons, including all coincident measurements, were possible. In this respect, it is to be specified that routine radiosonde launches are available from IMAA-CNR only starting from October 2014, so inter-comparisons before this date were very infrequent. Figure 2 illustrates the vertical profiles of the water vapour mixing ratio and temperature mean BIAS and RMS deviation for the 11 considered comparisons. The altitude interval 2.5-4 km was used to quantify the mean value of the calibration constant for water vapour measurements, $\overline{c}$, which is obtained by averaging the single calibration coefficient values from all 11 inter-comparisons. The uncertainty affecting the calibration constant, $\sigma_c$, has been estimated as the standard deviation all single calibration values from the mean value. The value of $\overline{c}$ is found to be equal to 82.33, while the value of $\sigma_c$ is found to be equal to 3.72. The standard deviation, expressed in percentage $(100 \times \sigma_c / \overline{c})$, is found to be equal to 4.5 %. A very similar procedure was applied to calibrate

temperature measurements. In this case the mean value and standard deviation of the calibration constants $a$ and $b$ were determined, with $\overline{a} \pm \sigma_a = 760 \pm 7$ and $\overline{b} \pm \sigma_b = 0.97 \pm 0.03$."

The constancy of the calibration coefficient has been verified over the two years measurement period reported in the paper. The calibration coefficient appears to be quite stable with time as in fact neither short-term nor long-term variations have been revealed. Ageing of transmitter/receiver components does not produce any appreciable variation of the calibration coefficients. In this regard it is to be specified that both water vapor mixing ratio and temperature measurements determined through the Raman technique are obtained by ratioing two Raman signals. These are the water vapour roto-vibrational Raman signal $P_{H2O}(z)$ and molecular nitrogen roto-vibrational Raman signal $P_{N2}(z)$ in the case of water mixing ratio measurements, while in the case of temperature measurements these are the low and high quantum number rotational Raman signal from $N_2$ and $O_2$ molecules, $P_{LoJ}(z)$ and $P_{HiJ}(z)$, respectively. $P_{H2O}(z)$ and $P_{N2}(z)$, as well as $P_{LoJ}(z)$ and $P_{HiJ}(z)$, are collected and detected with two channels which are very close one to the other in receiver, having most of the optical components in common (telescope primary and secondary mirrors, collimating optics and a variety of beam splitters). The only components that differ in the two receiving channels are the interference filters and the photomultipliers.

For what concerns the interference filters, aging - if occurring - would possibly affect both filters in a similar extent and consequently the effects would cancel out when ratioing the signals $P_{H2O}(z)$ and $P_{N2}(z)$ or $P_{LoJ}(z)$ and $P_{HiJ}(z)$. We specified "if occurring" as in fact aging effects on filters specifications over time intervals of 1-2 years are expected to be negligible and, to our knowledge, never reported in literature. Similar arguments are valid when considering the aging of the photomultipliers: if occurring, aging would possibly affect both photomultipliers, and consequently the effects would cancel out when ratioing the signals. Additionally, while photomultipliers' aging can potentially affect lidar signals (long-term operation of photomultipliers may lead to a depletion of the photocathode material), this effect is intrinsically related to their effective operation. In this regard, it is to be specified that the Raman lidar BASIL is operated only 4-8 hours per week, each Thursday evening, weather permitting. This translates into an overall photomultipliers' operation time of 300-400 hours over the two year period considered in the present paper, which is a very short period when compared to the real life-time of photomultipliers.

Furthermore, no evidence of environmental temperature effects on system performance has been observed. In this regard, it is to be specified that the system is hosted in a scientific sea-tainer and operated in a temperature and humidity controlled environment, so that temperature and humidity changes associated with the succession of seasons are not observed.

All the above mentioned aspects are now clearly specified in the text, where the following sentences have been introduced: "The constancy of the calibration constant was verified over the two years measurement period, appearing quite stable, as in fact neither short-term or long-term time variations were revealed. Ageing of transmitter/receiver components does not produce any appreciable variation of the calibration coefficients."

In the former version of the paper, we had introduced in a separate section (former section 6.5: "Assessment of the BIAS and RMS deviation: specific dedicated inter-comparison effort between BASIL and the radiosondes") all the comparison between the Raman lidar and the radiosondes from IMAA-CNR, which are the basis of the calibration. This section has now been removed and a large portion of its content has been ingested in the extended section now dedicated to the calibration (section 6.1 Raman lidar calibration).

- If the lidar has been calibrated compared with modelling data, is it not normal that the biases are small? AIRS, IASI, radiosonde and model data are not independent. AIRS and IASI operate on a similar way with average kernel leading to ~ 2x higher vertical resolution for IASI. Radiance data

are assimilated into the model, just like radiosonde data, so the reference profiles are not independent.

As already specified above, lidar data are not calibrated based on the comparison with modelling data, but with radiosondes. As specified above, a new section has now been introduced (new section 6.1 Raman lidar calibration) to illustrate the calibration procedure. However, we agree that the small mutual bias values between the different satellite sensors and the models are partially to be attributed to the fact that AIRS, IASI and model data are not independent. This is especially true for AIRS and IASI data, these two sensors being operated in a similar way, with their radiance measurements being analyzed with very similar algorithms and average kernels. Additionally, radiance data from these two space sensors are assimilated into the ECMWF and ECMWF-ERA model reanalysis, which make IASI/AIRS data and ECMWF model re-analyses is some extent mutually dependent. However, this is not true for the mutual biases between the radiosondes and the Raman lidar and between these two sensors and the satellite sensors and ECMWF model data. In fact, radiosondes from IMAA-CNR are not assimilated by ECMWF and the Raman lidar provides completely independent measurements, which are calibrated with unassimilated radiosonde data (this latter is from a research launching station and not from an operational station included in the upper air network). These aspects are now better clarified in the paper, where the following sentences have been introduced: "It is to be specified that IASI and AIRS data, together with a variety of additional sensors, are assimilated in ECMWF re-analyses, which makes ECMWF re-analyses partially dependent on IASI and AIRS data, with possible non-negligible effects on the mutual biases between the satellite and the model re-analyses data. However, the mutual biases between the radiosondes and the Raman lidar, and between these two sensors and the different satellite sensors and ECMWF re-analyses are completely unaffected by sensor/model cross-dependences, as in fact radiosondes from IMAA-CNR are not assimilated by ECMWF and the Raman lidar provides completely independent measurements, which are calibrated with unassimilated radiosonde data."

- How can such low biases be explained given that the profiles referred to are associated with much higher biases. For example, statistical studies on a large number of radiosondes have shown bias in the order of 0.4 g kg$^{-1}$ and 0.5 K. For IASI we are on 0.5-1 g kg$^{-1}$ and 0.5-1 K depending of the kernel averaging function.

As already anticipated above, the statement in the Abstract which refers to an accuracy, expressed in terms of bias, of 0.1 K for temperature measurements between the surface and 50 km and of 1 gkg$^{-1}$ for water vapour mixing ratio measurements between the surface and 15 km has not been properly justified and commented. Here we are referring to "measurement accuracy" and not "measurement bias", which has now been corrected in the text. The term "accuracy" is used to refer to the combined effect of both random and systematic errors on measurement performance. As properly illustrated above, the Raman lidar is calibrated against the radiosondes, considering an altitude region (2.5-4 km), where both sensors have high performance and are sampling the same air-masses. The measurement accuracy not exceeding 0.1 K and 0.1 g kg$^{-1}$, which we mention in the Abstract, is computed from the mutual bias and RMS deviation between BASIL and the radiosondes in the upper troposphere region, whose mean value is not exceeding 0.1 g kg$^{-1}$ for water vapour mixing ratio measurements and 0.1 K for temperature measurements.

Furthermore, the low bias values reported in the second paragraph of the abstract are those of the vertically-averaged mean bias, which indeed is a quantity having values smaller than the single bias values at different altitudes. In the revised version of the paper, in addition to the vertically-averaged mean bias, $\overline{bias}$, i.e. expression (17) in the paper, originally used by Bhawar *et al.* (2011), we decided to also introduce the vertically-averaged absolute bias, $\overline{|bias|}$, defined as the weighted mean of the absolute values (this time in the mathematical sense) or moduli of the single bias values

at different altitudes (see above). Values of $\overline{|bias|}$ are found to be closer to those referred by the referee.

These aspects are now better specified in the text, where the following new sentences have been introduced: "So far we have reported and discussed the mutual bias and RMS deviation profiles between different sensors/models, highlighting the altitude variability of these quantities. However, in order to assess sensors and models performance is often preferable to use a single bias/RMS deviation value. This leads us to the definition of the vertically-averaged mean bias and the vertically-averaged absolute mean bias."

Further down in the text, we also introduced the following new sentences: "The vertically-averaged absolute mean bias, $\overline{|bias|}$, and RMS deviation, defined as the weighted mean of the moduli of the single biases at different altitudes, can be determined through the expression:

$$\overline{|bias/RMS|} = \frac{\sum_{i=1}^{N} w_i |bias_i/RMS_i|}{\sum_{i=1}^{N} w_i}$$

(18)

In the vertically-averaged absolute mean bias $\overline{|bias|}$, values at different altitudes with different signs will not cancel out. Consequently, values of $\overline{|bias|}$ are larger than the corresponding $\overline{bias}$ values."

- Why limit yourself to 4 case studies? This considerably limits the investigation and makes the statistical study unrepresentative. The argument developed at the beginning of section 6.1 is not relevant.

The reported measurements cover a ~ 1.5 year period (17 months from 7 November 2013 to 9 April 2015). In the previous version of the paper we concentrated our attention on four case studies, which had been analyzed with a customized approach, instead of considering a larger dataset analyzed with a standard routine analysis approach. Such an approach had been considered for the purpose of minimizing the effects on the statistical analysis associated with the application of a routine data analysis approach. In the revised version of the manuscript we are now including another 2 case studies, for a total of 6 case studies. The same customized analysis approach has been applied also to these two additional cases. It is to be specified that this represents the complete data set of clear air case studies. In fact, clear sky condition is the most suited condition to perform both water vapour and temperature measurements by Raman lidar, with water vapour profile measurements extending up to the UTLS region and temperature profile measurements extending up to 50 km. In this regard it is to be highlighted that an appropriate assessment of measurement performance based on a sensors/models inter-comparison effort requires all sensors to be operated in clear sky conditions, which is not always the case for either the Raman lidar or the two passive space sensors IASI and AIRS. More specifically, the Raman lidar system BASIL does not have an all-weather measurement capability, which implies that the system is shut down in case of precipitation. Additionally, BASIL (and this is true for all lidar systems) cannot penetrate thick clouds, the laser beam being completely extinguished for optical thicknesses around 2. Acceptable Raman lidar performance are still possible above thin clouds, with optical thickness < 0.3. This translates into the fact that, for the purposes of the present inter-comparison effort, even the presence of high cirrus clouds makes case studies non eligible for the comparison. In two other specific case studies IASI and/or AIRS data were characterized by a very poor quality and unrealistic biases, which forced us to remove those from the inter-comparison effort. All in all, the overall number of left possible inter-comparisons is 6 and the statistical analysis was re-run again of these 6 case studies. This aspect has been now clarified in the text, where the corresponding paragraph reads as follows: "For the aims of this paper, we focused our attention on six selected

case studies collected during the first 2 years of operation of the system, namely 7 November 2013, 19 December 2013, 9 October 2014, 27 November 2014, and 2 and 9 April 2015. While a larger data-set could have been chosen, we decided to focus our attention only on clear sky cases. In fact, clear sky condition represents the most suited condition to perform both water vapour and temperature measurements by Raman lidar, with water vapour profile measurements extending up to the UTLS region and temperature profile measurements extending up to 50 km. An appropriate assessment of measurement performance based on a sensors/models inter-comparison effort requires the sensors to be operated in clear sky conditions, which is not always the case for either the Raman lidar or the two passive space sensors IASI and AIRS. More specifically, the Raman lidar system BASIL does not have an all-weather measurement capability, which implies that the system is shut down in case of precipitation. Additionally, BASIL - and this is true for all lidar systems - cannot penetrate thick clouds, the laser beam being completely extinguished for optical thicknesses around 2. Acceptable Raman lidar performance are still possible above thin clouds, with optical thickness < 0.3. Thus, for the purposes of the present inter-comparison effort, even the presence of high cirrus clouds makes case studies non eligible for the comparison. In other case studies IASI and/or AIRS data were characterized by a very poor quality and unrealistic biases, which forced us to remove those from the inter-comparison effort."

- The BASIL lidar has integrated the NDACC network which already contains other water vapour and temperature lidars. It would therefore relevant to compare the accuracy of these different lidars with that of BASIL.

This is a very good suggestion and we have now introduced two paragraphs based on literature results aimed at comparing the accuracy of BASIL with that of other Raman lidars integrated in the NDACC network. More specifically, we are now comparing the water vapour performance of BASIL with the NASA-JPL Raman lidar operated at Table Mountain (USA, Leblanc et al., 2012), with the ALVICE Raman lidar operated at NASA Goddard Space Flight Center (Whiteman et al., 2012) and with the CNRS Raman lidar operated at Maïdo Facility (Réunion island, Dionisi et al., 2015). Additionally, we are also comparing the temperature performance of BASIL with the NASA-JPL Raman lidar operated at Table Mountain (USA, Leblanc et al., 1998b) and with the Rayleigh lidar in Thule (Marenco et al., 1997).

For what concerns water vapour measurements, Whiteman et al. (2012) reported a 5% uncertainty in the upper troposphere based on an extended comparison of the Raman lidar system ALVICE in NASA-GSFC with Vaisala RS92 radiosondes. Dionisi et al., 2015 reported a relative difference below 10 % in the low and middle troposphere (2–10 km) for the Maïdo Lidar in Réunion island based on the comparison with 15 co-located and simultaneous Vaisala RS92 radiosondes. The upper troposphere, up to 15 km, is found to be characterized by a larger spread (approximately 20 %), attributed to the increasing distance between the two sensors.

[revised manuscript text omitted]

- To improve the clarity of the study, 2 sets of profiles should be defined. The first set would be dedicated to calibration and the second to the study of biases and standard deviations.

This is actually the way we have reformulated the text of the manuscript. We are now first introducing the calibration procedure applied to the Raman lidar water vapour mixing ratio and temperature measurements (section 6.1 Raman lidar calibration). The calibration procedure is based

on the comparison of Raman lidar profiles with simultaneous radiosonding data. Two subsequent separate sections (6.2 Case studies and 6.3 Assessment of the BIAS and RMS deviation between the different sensors/models) are now dedicated to the illustration of the inter-comparison results in terms of biases and root-mean-square deviations. We thank the reviewer for this precious suggestion which makes the manuscript clearer and easier to read.

The authors place their work so heavily that they forget the existence of works conducted by other teams around the world. There were other NDACC publications, other cross-comparison exercises using Raman technology, very extensive studies on the representativeness of radiosondings and comparisons to modelling.

We agree with the reviewer that a number of citations of the scientific work conducted by other research teams are missing and we apologize for this lack of cross-references. As already mentioned above, we are now explicitly citing a variety of additional NDACC publications assessing the lidar performances in terms of water vapour mixing ratio (Leblanc et al., 2012; Whiteman et al., 2012; Dionisi et al., 2015) and temperature measurements (Marenco et al., 1997; Leblanc et al,. 1998b; Dou et al., 2009) based on inter-comparison exercises and very extensive studies of radiosondings data representativeness and comparisons with modelling.

Specific comments
Abstract. It must be revised after taking into account the previous remarks.

The Abstract has been revised taking into account the previous remarks on measurement accuracy. Specifically, the fourth sentence in the Abstract has been changed as follows: "Measurements illustrated in this manuscript demonstrate the ability of BASIL to perform measurements of the temperature profile up to 50 km and of the water vapour mixing ratio profile up to 15 km, when considering an integration time of 2 h and a vertical resolution of 150-600 m, with measurement mean accuracy, determined based on comparisons with simultaneous and co-located radiosondes, of 0.1 K and 0.1 g kg$^{-1}$, respectively, up to the upper troposphere. Relative humidity profiling capability up to the tropopause is also demonstrated by combining simultaneous temperature and water vapour profile measurements." Additionally, the second paragraph of the Abstract has been completely rewritten and now reads: "Raman lidar measurements are compared with measurements from additional instruments, such as radiosondings and satellite sensors (IASI and AIRS), and with model re-analyses data (ECMWF and ECMWF-ERA). We focused our attention on four selected case studies collected during the first 2 years of operation of the system (November 2013-October 2015). Comparisons between BASIL and the different sensor/model data in terms of water vapour mixing ratio indicate biases in the altitude interval 2-15 km always within the interval ± 1 g kg$^{-1}$ (or ± 50 %), with minimum values being observed in the comparison of BASIL vs. radisondings (± 50 % up to 15 km). Results also indicate a vertically-averaged mean mutual bias of -0.026 g kg$^{-1}$(or -3.8 %), 0.263 g kg$^{-1}$ (or 30.0 %), 0.361 g kg$^{-1}$(or 23.5 %), -0.297 g kg$^{-1}$ (or -25 %), -0.296 g kg$^{-1}$ (or -29.6 %), when comparing BASIL versus radisondings, IASI, AIRS, ECMWF, ECMWF-ERA, respectively. Vertically-averaged absolute mean mutual biases are somewhat larger, i.e. 0.05 g kg$^{-1}$(or 16.7 %), 0.39 g kg$^{-1}$ (or 23.0 %), 0.57 g kg$^{-1}$(or 23.5 %), 0.32 g kg$^{-1}$ (or 29.6 %), 0.52 g kg$^{-1}$ (or 53.3 %), when comparing BASIL versus radisondings, IASI, AIRS, ECMWF, ECMWF-ERA, respectively. For what concerns the comparisons in terms of temperature measurements, results indicate mutual biases in the altitude interval 3-30 km always within the interval ± 3 K, with minimum values being observed in the comparison of BASIL vs. radisondings (± 2 K within the same altitude interval). Results also reveal mutual biases within ± 3 K up to 50 km for most sensor/model pairs. Results also indicate a vertically-averaged mean mutual bias between BASIL and the radisondings, IASI, AIRS, ECMWF, ECMWF-ERA of -0.03, 0.21, 1.95, 0.14, 0.43 K, respectively. Vertically-averaged absolute mean mutual biases between BASIL and the radisondings, IASI, AIRS, ECMWF, ECMWF-ERA are 1.28, 1.30, 3.50, 1.76, 1.63 K, respectively.

Based on the available dataset and benefiting from the circumstance that the Raman lidar BASIL could be compared with all other sensor/model data, it was possible to estimate the overall bias of all sensors/datasets, this being 0.004 g kg$^{-1}$/0.30 K, 0.021 g kg$^{-1}$/-0.34 K, -0.35 g kg$^{-1}$/0.18 K, -0.346 g kg$^{-1}$/-1.63 K, 0.293 g kg$^{-1}$/-0.16 K  and 0.377 g kg$^{-1}$/0.32 K for the water vapour mixing ratio/temperature profile measurements carried out by BASIL, the radiosondings, IASI, AIRS, ECMWF, ECMWF-ERA, respectively."

Repetitions should be avoided (L17).

The sentence which was replicating the information on the altitude region for the water vapour mixing ratio and temperature comparisons has been removed.

The end of the abstract is reminiscent of circular reasoning.

We are not sure we understand what is the circular reasoning the referee is referring to. However, we certainly generated some confusion not properly specifying from the very beginning what we mean for mutual and overall bias. The main difference is between "overall bias" and "mutual bias". The mutual bias is the deviation between two profiles from a pair of sensors/models, with all possible pairs being: BASIL vs. radiosondings (RS), BASIL vs. IASI, BASIL vs. AIRS, BASIL vs. ECMWF, BASIL vs. ECMWF-ERA, RS vs. IASI, RS vs. AIRS, RS vs. ECMWF, RS vs. ECMWF-ERA, IASI vs. AIRS, IASI vs. ECMWF, IASI vs. ECMWF-ERA, AIRS vs. ECMWF, AIRS vs. ECMWF-ERA and ECMWF vs. ECMWF-ERA.

We have now better specified that in the first part of the second paragraph of the Abstract we refer to the mutual bias between BASIL and all other sensors/models (BASIL vs. RS, BASIL vs. IASI, BASIL vs. AIRS, BASIL vs. ECMWF, BASIL vs. ECMWF-ERA), while in the final part of the second paragraph we refer to the "overall bias" affecting each sensor/model. The overall bias affecting each sensor/model can be determined benefiting from the circumstance that BASIL could be compared with all other sensor/model data.

The overall bias is the deviation between one sensor/model profile and the reference profile. As no reference sensor/model was available, we considered as reference profile the average profile of all sensors/models involved in the inter-comparison effort. The estimate of the overall bias, as clearly explained in section 6.4, is obtained by imposing that the summation of all mutual biases between sensor/model pairs is equal to zero, assuming equal weight for the data reliability of each sensor/model. The choice of attributing equal weight to the data reliability of each sensor/model is driven by the awareness that none of them can *a priori* be assumed more accurate than the others and consequently a "pseudo" reference profile can be obtained by averaging all profiles.

In the previous version of the paper we had also distinguished between relative and absolute biases, using the term "absolute" to refer to the bias having the same units of the atmospheric quantity, i.e. being a physical quantity homogeneous to the atmospheric quantity (g kg$^{-1}$ for water vapour mixing ratio and K for temperature), while relative bias refers to the percentage bias, i.e. the absolute bias divided for the atmospheric quantity and multiplied for 100. In the revised version of the paper, for the purpose of avoiding any misunderstanding, the have avoided to use the terms "absolute bias" and "relative bias" in the sense specified above and we now more easily refer to "bias" and "percentage bias", respectively.

Can an absolute bias be negative?

As explained above, and now also properly specified in the text, in the previous version of the paper the absolute bias was intended to indicate the bias having the same physical units of the considered atmospheric quantity, i.e. being a physical quantity homogeneous to the atmospheric quantity, and it was not intended as the absolute value in mathematical terms. In order to avoid misunderstandings, we have now redefined the considered quantities and we are now using the terms "bias" and "percentage bias" to substitute the previously used terms of "absolute bias" and "relative bias",

respectively. The referee asks if an absolute bias can be negative. If we consider the meaning for "absolute" used in the former version of the paper (i.e. a quantity with the same physical units of the measured parameter), it is indeed possible, as in fact we are not considering the absolute value in mathematical terms (i.e. the modulus |x| of a real number x, which is the non-negative value of x without regard to its sign).

Introduction P2L21. The cost of a radiosonding is about 250 € Is a lidar competitive at this level, especially for the upper troposphere and stratosphere?

The cost of a lidar system and its operation is competitive with the cost of a radiosonding system and its operation. In fact, the cost of the radiosonde launching system is comparable with the cost of the research lidar facility (200-250 k€), while the operational cost is much lower for the Raman lidar than for the radiosonde: each radiosonde launch has a cost of ~250 €, while the cost of operating the lidar system over a 2 hour time interval (i.e. the integration time needed to obtain a sufficiently high signal statistics to obtain high precision measurements of the water vapour mixing ratio profile up to 15 km and of the temperature profile up to 30 km, this latter being the altitude region typically covered by radiosondes before blasting) is of the order of 18-20 € More specifically, with a pulse repetition rate of 20 Hz, the 2 hour integration time correspond to 144.000 laser shots. The most short-lived and perishable components of flash lamp-pumped solid state laser sources used for lidar purposes are the flash-lamps themselves, which are used to optically pump the lasing material. Flash-lamps usually have a life time of 40-50 million shots, each lamp costing approx. 1000 € with two of them being typically needed in a master-oscillator power-amplifier laser configuration. The cost in terms of flash-lamps for a 2 hour integration time corresponds to 11-14 € (18-20 € in the worst case scenario when considering accidental deterioration of other optical components), which is the amount to be compared with the 250 € cost of a radiosonde launch. Thus, the operation of the Raman lidar is at least an order of magnitude cheaper that the use of radiosondes. In addition to being more expensive than the Raman lidar, radiosonding cannot be launched with a schedule sufficiently intense to guarantee the temporal resolution needed for some of the scientific scopes identified in the paper (for example, meteorological process studies).

P3L5. Perhaps there is an overlap factor?

The use of a very compact optical design reduces significantly the differences between the overlap functions of the $H_2O$ and $N_2$ Raman signals, $P_{H2O}(z)$ and $P_{N2}(z)$, used to estimate the water vapor mixing ratio profile. This is also true for the overlap functions of the low J and high J pure rotational Raman lidar signals, $P_{LoJ}(z)$ and $P_{HiJ}(z)$, which are used to estimate the temperature profile. Having almost identical overlap functions for the two signals that are ratioed ($P_{H2O}(z)$ over $P_{N2}(z)$ and $P_{LoJ}(z)$ over $P_{HiJ}(z)$)) allows to extend the water vapor mixing ratio and temperature profile measurements down to the proximity of the surface, with the only exclusion of the so called "blind region", typically the lowest 100-150 m. Consequently, the temperature profile measurements by BASIL based on the pure-rotational Raman technique cover the altitude interval from almost surface up to 20-25 km. This aspect is now clearly specified in the text, where the following sentence has been introduced: "The possibility to measure down to the proximity of the surface is guarantee by the very compact optical design of the lidar receiver, which translates into negligible differences between the overlap functions of the two ratioed Raman signals (see details in section 4.2.1)".

In section 4.2.1 the following text has also been introduced: "The use of a very compact optical design for the lidar receiver reduces significantly the differences between the overlap functions of the roto-vibrational Raman signals $P_{H2O}(z)$ and $P_{N2}(z)$ used to determine the water vapor mixing ratio profile, as well as the differences between the overlap functions of the pure-rotational Raman signals $P_{LoJ}(z)$ and $P_{HiJ}(z)$) used to determine the temperature profile. This translates into the capability for the present system to extend water vapor mixing ratio and temperature profile

measurement down to the proximity of the surface, with a marginal blind region corresponding to the lowest 100-150 m".

P3L10-11. Not necessary because it is not a paper topic.

The sentence in page 3, lines 10-11, has now been removed.

Section 2 It would be interesting to have a table that summarizes the main characteristics of the lidar.

A table summarizing the main characteristics of the lidar system has been introduced (new table 1), which follows below together with its table caption.

| **Laser** | Nd:YAG |
|---|---|
| Wavelengths | 354.7, 532 nm |
| Single pulse energy | 500 mJ @ 354.7 nm, 300 mJ @ 532 nm |
| Pulse repetition frequency | 20 Hz |
| Beam divergence | 0.5 mrad (FWHM) |
| **Telescope** | Newtonian configuration |
| Primary mirror diameter | 0.45 m |
| Combined focal length | 1.8 m |
| Field of view | 0.5 mrad (FWHM) |
| **Interference filters** | Elastic, $N_2$, $H_2O$, LoJ, HiJ |
| Center wavelength (nm) | 354.7, 532, 386.7, 407.5, 354.3, 352.9 |
| Bandwidth (nm) | 1.0, 1.0, 1.0, 0.25, 0.2, 1.0 |
| Blocking at 354.7 nm | -, $10^{-6}$, $10^{-10}$, $10^{-12}$, $10^{-8}1$, $10^{-8}$ |

Table 1: Main characteristics of the Raman lidar system BASIL.

Former tables 1 and 2 have now been renamed 2 and 3, respectively.

P3L28 and 29. Typography

We are not sure what the reviewer refers to with the term "Typography" here. If it refers to the typographical error associated with the *double dot* after the term "approx", this typing error has now been removed.

Section 3.1 P5L5. There are much earlier references.

As suggested by the reviewer, in order to properly introduce the Infrared Atmospheric Sounding Interferometer (IASI), we are now citing two additional much earlier papers: Siméoni et al., 1997 and Rabier et al., 2002. At the same time, we removed the late citation by Masiello et al. (2013).

P5L12. For numerical weather forecasting, IASI inversions are not used, radiances are assimilated directly.

As properly pointed out by the reviewer, IASI inversions in terms of atmospheric parameters are not assimilated in numerical weather forecasting, while IASI radiances are directly assimilated. This aspect has now been clearly specified in the text, where the corresponding sentence has been changed as follows: "Such performance may have a major impact on many scientific areas, especially on Numerical Weather Prediction, where at present only IASI radiances are directly assimilated".

Sections 4.1, à 4.3 These sub-sections are already very well known, it is enough to highlight the sources in order to simplify the article.

Sections 4.1 through 4.3 have been substantially shortened. Besides the calibration paragraph, that has been extended in accordance with the requests of the reviewer, only three sentences are left in section 4.1. Section 4.2 has been shortened by more than one page. Section 4.3 is now only three sentence long and could not be shortened any further.

The important point is the calibration which needs to be clarified.

As requested by the reviewer, the calibration procedure has now been extensively clarified. One new sub-section (6.1 Raman lidar calibration) has been introduced (with a length exceeding 1 page), where all the aspects concerning the determination of the calibration constants and the assessment of their time stability are extensively addressed.

The choice of a two-parameter temperature adjustment function must be justified. This type of adjustment does not guarantee optimal accuracy, it is preferable to use functions with 3 parameters, especially with a wide temperature range. It is surprising to obtain such low biases with this type of function.

The choice of a two-parameter temperature calibration function is motivated based on the consideration of the number of rotational lines actually selected by the interference filters and exploited for the temperature measurements. In fact, the two-parameter calibration function is exactly valid for two individual lines (Arshinov et al., 1983), but it is also valid when a very limited number of rotational lines are selected in both the low J and high J portions of the pure-rotational Raman spectrum of $N_2$ and $O_2$ molecules, as is the case for the spectral selection configuration implemented for BASIL. Here, 3 lines are selected in the low J spectral region and 6 lines are selected in the high J spectral region. In this case, the use of a two-parameter temperature calibration function leads to a systematic error typically not exceeding 0.2 K (Di Girolamo et al., 2006). However, in case a larger number of rotational lines are selected both in the low J and high J portions of the pure-rotational Raman spectrum, improvements in temperature measurements can be obtained by introducing higher-order terms in the temperature calibration function (Behrendt, 2005). This aspect is now better clarified in the paper, where the following sentence has been introduced: "In the case of BASIL, a two-parameter calibration function is well suited for the determination of the temperature profile from the $P_{LoJ}(z)$ and $P_{HiJ}(z)$ as in fact a limited number of rotational lines are selected for this purpose both in the low J and high J portions of the pure-rotational Raman spectrum (Di Girolamo et al., 2006)".

The method developed by Auchecorne et al. is already well described and the error sources have been seriously studied and evaluated. Instead, errors should be discussed in this section because the results are very dependent on the lidar used (e.g. the optical filtering technique used). The methods developed by the pioneering authors are to be considered but applied to the BASIL lidar. There is no reason to achieve the same levels of error.

Systematic errors affecting temperature measurements by BASIL based on the application of the lidar integration technique had not been assessed in the previous version of the paper and an estimate had been provided only based on literature values. In the revised version of the paper we are now carefully assessing the systematic uncertainty affecting temperature measurements by BASIL when considering an upper reference altitude $z_{ref,2}$ and applying the algorithm downward. Elastic signals from BASIL extend with sufficiently high statistics up to approx. 55 km. Consequently, $z_{ref,2}$ is taken equal to 55 km and boundary values $T(z_{ref,2})$ and $N(z_{ref,2})$, to be known with sufficient accuracy, are taken from the mid-latitude reference models from the U.S. Standard Atmosphere (1976), considering the different seasonal options included therein (Kantor and Cole, 1962). The algorithm is applied downward, initializing at 55 km. Although the boundary value of $T(z_{ref,2})$ taken from the model atmosphere may differ from the real value, the systematic uncertainty affecting the measurement at an altitude of 5 km below $z_{ref,2}$, i.e. at 50 km, is smaller than 1 K. This is clearly demonstrated by the results reported in the paper, revealing deviations at this altitude between BASIL and the model re-analyses ECMWF and ECMWF-ERA smaller than 1 K for all

case studies, this value (1 K) being considerably smaller than the statistical uncertainty affecting temperature measurements from BASIL at this same altitude (± 2 K). This aspect is now clearly specified in the paper, where the following sentences have been introduced: "Similar considerations are also valid for BASIL. In this case, the elastic signals extend with a sufficiently high statistics up to approx. 55 km; thus, $z_{ref,2}$ is taken equal to 55 km and boundary values $T(z_{ref,2})$ and $N(z_{ref,2})$ are taken from the mid-latitude reference atmospheric models of U.S. Standard Atmosphere (1976), considering the different seasonal options included therein (Kantor and Cole, 1962). The systematic uncertainty affecting the measurement at an altitude of 5 km below $z_{ref,2}$, i.e. at 50 km, is smaller than 1 K, as clearly highlighted by the results reported in sections 6.1 and 6.2, which reveal deviations at this altitude between BASIL and model re-analyses ECMWF and ECMWF-ERA smaller than 1 K for the case studies, i.e. considerably smaller than the statistical uncertainty affecting BASIL temperature measurements at this altitude (± 2 K)."

In the revised version of the paper we are now also reporting the systematic uncertainties due to considering an upper reference altitude $z_{ref,2}$, which are affecting temperature measurements by a variety of other Rayleigh lidar systems included in NDACC. In this direction, the following sentences have been introduced: "The bias values listed above are in agreement with those reported for a variety of other Reileigh lidars operated in the frame of NDACC. Specifically, Marenco et al. (1977) reported for the Rayleigh lidar in Thule (Greenland) a potential systematic uncertainty, or bias, associated with the selection of incorrect upper boundary values smaller than the statistical uncertainty affecting the measurements (± 2 K). These results were obtained based on a dedicated sensitivity analysis, with upper boundary values varied by 5 %. Leblanc et al. (1998b) reported bias values from a variety of temperature lidar systems based on Rayleigh technique included in NDACC. Specifically, temperature measurements from the CNRS-SA Rayleigh lidars at Observatoire de Haute Provence (France) and at the Centre d'Essais des Landes were found to be characterized by a bias smaller than 1 K at 55 km, while those from the NASA-Jet Propulsion Laboratory Rayleigh lidars located at Table Mountain (California) and at Mauna Loa (Hawaii) were characterized by a bias smaller than 1 K at 55 and 50 km, respectively. A bias of ~1 and ~2 K, again associated with the selection of incorrect upper boundary values, was found to characterize the Rayleigh lidars located at Hohenpeissenberg (Germany) and Sondre Stromfjord (Greenland), respectively (Dou et al., 2009).".

Section 4.2.2 P9L20. How is this altitude interval justified? This is a critical point to reduce the uncertainty random and it must be justified for each lidar.

Here, in order to reduce the random uncertainty affecting the estimate of the calibration constant $C$, we consider an average over 10 data points, i.e. 5 data points below and 5 above the reference altitude $zref,1$ (typically around 20 km). When carrying out density measurements by lidar using a vertical resolution of 150 m, the aerosol- and cloud-free atmospheric region used for the calibration is 1500 m. Averaging over 10 data points leads to an estimate of the calibration coefficient with an uncertainty which is much smaller (in excess of a factor of 3) than the statistical uncertainty affecting the single number density profile values. However, this portion of the text is no longer present in the paper as in fact, in order to cope with a previous request from the same reviewer ("Sections 4.1, à 4.3 These sub-sections are already very well known, it is enough to highlight the sources in order to simplify the article."), section 4.2 has been sensitively shortened - by more than one page - and the description of the technique considered for atmospheric number density profile measurements and their calibration have been removed, citing instead specific literature papers.

P10L6. Maybe a little less, 80 km on average?

The upper altitude of the homosphere has now been changed into 80 km.

P10L10. It is not the good term

The term here has been changed from "progressively extrapolating …" to "progressively applying …"

P10L23. It is mainly due to the stability of the equation which limits the error propagation.

Indeed, when an upper reference altitude is taken and the algorithm is applied downward, uncertainties affecting the assumed values of $T(z_{ref,2})$ and $N(z_{ref,2})$ do not affect the temperature profile $T(z)$ few kilometers below the reference altitude and the systematic uncertainty affecting $T(z)$ quickly reduces (Behrendt, 2005), mainly due to the stability of the equation which limits the error propagation. This aspect is now properly specified in the text, where the corresponding sentence has been changed as follows: "On the contrary, when an upper reference altitude is taken and the algorithm is applied downward, errors affecting $T(z_{ref,2})$ and $N(z_{ref,2})$ are not affecting the temperature profile $T(z)$ few kilometers below the reference altitude, with the systematic uncertainty affecting $T(z)$ quickly reducing (Behrendt, 2005), mainly due to the stability of the equation which limits the error propagation".

Equation 16. Subscripts are missing

We have now properly introduced the subscripts in this equation.

Equation 17. Idem.

We have now properly introduced the subscripts also in this equation.

Are you sure about this relative RMS calculation? You do a simple average on RMS.

We are sorry, but there was a typing error in equation 17, now equation 15. The term in square brackets under the root sign has to be squared and should read:

$$RMS = \frac{1}{N} \sum_{i=1}^{N} RMS_i = \frac{2}{N} \sum_{i=1}^{N} \left\{ \frac{\sqrt{N \sum_{z=z_1}^{z_2} [q_1(z) - q_2(z)]^2}}{\sum_{z=z_1}^{z_2} [q_1(z) + q_2(z)]} \right\}$$

P12L12. What type of interpolation?

Water vapour mixing ratio and temperature data are interpolated through a linear function. This is now clearly specified in the text, where the corresponding sentence has been modified as follow: "A linear interpolation is used in the present effort for the water vapour mixing ratio and temperature data."

P12L13. The bias is signed, so it is not absolute.

As we already mentioned above, in this paper, in compliance we the definitions used in previous papers (Whiteman et al. 2006, Behrendt et al., 2007a, Behrendt et al., 2007b, Bhawar et al. 2011), with the term "absolute bias" and "absolute RMS deviation" we were intending the bias and RMS values having the same physical units of considered atmospheric quantity, i.e. being physical quantities homogeneous to the atmospheric quantities, and they were not intended as the absolute values in mathematical terms (in mathematical terms, the absolute value or modulus |x| of a real number x is the non-negative value of x without regard to its sign). Conversely, with the term "relative bias" and "relative RMS deviation" we were intending the "percentage bias" and "percentage RMS deviation", i.e. the "absolute bias" and "absolute RMS deviation" divided for the mean value of the atmospheric quantity and multiplied for 100. In the revised version of the paper, for the purpose of avoiding any misunderstanding, the have avoided to use the terms "absolute bias" and "relative bias" in the sense specified above and we now more easily refer to "bias" and "percentage bias", respectively.

P13L14-15. Already explained above.

The reviewer is right: the presence of four selected case studies, although with less detail, had already been specified in the last sentence of the previous paragraph. This previous sentence has now been removed.

The first paragraph of section 6.1 should be put before.

The first paragraph of former sub-section 6.1, containing general information on the selected case studies, has now been anticipated and incorporated in the introductory part of Section 6, (Inter-comparison results). New section 6.1 is now dedicated to the description of the calibration procedure for the Raman lidar.

P14L6. How are such values calculated?

The sensitivity level of 0.001-0.002 g kg$^{-1}$ has been identified as the mixing ratio value corresponding to 100 % percentage (or relative) uncertainty in the UTLS region, while the measurement capability level of 0.003-0.004 g kg$^{-1}$ is identified as the mixing ratio value corresponding to a percentage (or relative uncertainty) of 50 % in this same altitude region. This aspect is now better specified in the paper, where corresponding sentence has been changed as follows: "The water vapour mixing ratio profile from BASIL reaches an altitude of approx. 15 km, with the capability to measure humidity levels as small as 0.003-0.004 g kg$^{-1}$, with a sensitivity level of 0.001-0.002 g kg$^{-1}$, the two levels being defined as the mixing ratio values corresponding to 50 and 100 % relative uncertainty in the UTLS region".

P14L21. What type of numerical filter?

The vertical smoothing filter we are using here is a simple moving or running average. More specifically, this is a central moving average computed using equally spaced data on either side of the point in the series where the mean is calculated and requires using an odd number of data points in the filter window. This aspect is now clearly specified in the text, where corresponding sentence has been changed as follows: "The considered vertical smoothing filter is a simple central moving or running average computed using equally spaced data (vertical step=30 m) on either side of the point where the mean is calculated, which requires using an odd number of data points in the filter window".

P21L9. There have already been cross-comparison studies of IASI and Raman lidar that reveal such differences in the ABL; differences due to the characteristics of IASI. Maybe you should talk about it.

As suggested by the reviewer, we are now citing previous cross-comparison studies of IASI and Raman lidar revealing differences in the ABL. Specifically, we are now citing the paper by Chazette et al. (2014), who revealed discrepancies between IASI and the Raman lidar in the planetary boundary layer, which the authors attribute to the weighting functions of IASI not being able to correctly sample the layer close to the ground. This aspect is now clearly specified in the text, where the following sentence has been introduced: "This missing capability of IASI to properly reproduce water vapour structures within the boundary layer had already been reported by Chazette et al. (2014), based on an extensive comparison of Raman lidar and IASI profile measurements carried out in the frame of the HyMeX and ChArMEx programs, attributing it to the weighting functions of IASI not correctly sampling layers close to the ground".

Section 6.3. This paragraph is cumbersome to read and could be greatly reduced with a better synthesis.

The paragraph was substantially shortened (by approx. half a page) and now reads much better.

Acknowledgments Many acknowledgments are missing for the data sets used, whether space-borne missions or modelling.

We are now acknowledging a variety of institutions for the provision of space measurements and modelling data. In this regard, the following sentences have been introduced: "ECMWF data used in this study have been obtained from the ECMWF Data Server (https://apps.ecmwf.int/datasets/data/interim-full-daily/levtype=sfc/). IASI Level-2 wter vapor mixing ratio and temperature profiles used in this paper are taken from EUMESAT database (https://eoportal.eumetsat.int/userMgmt/protected/ welcome.faces), while AIRS data are obtained from the NASA Goddard Earth Sciences Data Information and Services Center (GESDISC)."

**Anonymous Referee #2**

The paper presents a thorough assessment of the performance of the Raman lidar BASIL operated in Potenza, through a series of intercomparison with a variety of sensors (in situ and passive remote) and numerical weather prediction models. This commendable effort is conducted on the BASIL dataset acquired since November 2013, i.e. after BASIL has integrated the NDACC network. The paper focuses on comparisons with q et T profiles from radiosoundings launched from a nearby met station, as well as q and T profiles derived from the IASI and AIRS sensors in the vicinity of Potenza. Comparison with q and T profiles from 2 analyses from the ECMWF NWP model are also included. The results contained in this paper are of interest to the NDACC community. Nevertheless, the paper is tedious to read, which may in part be related to the fact that the authors present several inter-comparison periods (4 case studies and 2 longer periods) for which biases, deviations are discussed at length in both absolute and relative values.

In the modified version of the paper only one extended period is now illustrated. This allowed to severely shorten section 6.2 (Case Studies). All other case studies are now illustrated together in the same paper section only in terms of biases and RMS deviations (6.3 Assessment of the BIAS and RMS deviation between the different sensors/models), thus substantially shortening the discussion of biases and percentage biases and making the manuscript easier and less tedious to read.

My understanding is that the comparison with the radiosounding data from the station nearby Potenza is the key for a proper calibration of the Raman system. I think that the authors should start presenting this aspect thoroughly first before declining the comparison in the framework of 4 case studies.

The Raman lidar has been calibrated based on comparison with radiosondes and for this purpose we considered an extensive inter-comparison effort, making use of the radiosondes launched from the nearby station of IMAA-CNR. This is now clearly specified in the text, where a completely new section (section 6.1 Raman lidar calibration) – one page length - has been introduced to illustrate the calibration procedure. The results of the inter-comparison effort are now illustrated in following separate sections (6.2 and 6.3).

However, I am under the impression that radiosoundings may not be the only datasets used to "calibrate" the Raman retrievals (e.g. the work conducted for the case study on 7 November 2013 for which the authors state that there were no radiosounding data available). I would encourage the authors to clarify this in the revised manuscript. Are they using reanalysis products for calibration of BASIL?

The Raman lidar was calibrated only based on the use of the radiosoundings launched from the nearby station of IMAA-CNR. As already mentioned in response to referee # 1, an extensive inter-comparison effort was considered for this purpose based on the use of the radiosondes launched from the nearby station of IMAA-CNR (now extensively described in section 6.1 Raman lidar calibration). In this respect it is to be specified that this was the most appropriate approach we could actually consider as in fact the radiosonde launching station is only 8.2 km away from the Raman lidar and this gives confidence of the possibility to compare the Raman lidar and radiosonde profiles above the boundary layer. In this regard, it should be pointed out that in clear sky

conditions the horizontal homogeneity of the humidity field above the boundary layer top is sufficiently high to allow assuming that the two systems (the Raman lidar and the radiosonde) are sounding the same air masses. More specifically, for the purpose of calibrating the Raman lidar we compared Raman lidar and radiosonde profiles over the altitude interval 2.5-4 km. Within this altitude interval, while we assume water vapour heterogeneity to be small, we also can rely on high Raman lidar signals' strength, and consequently high signal-to-noise ratio levels and small statistical uncertainties. At the same time, within this low level altitude interval, the radiosonde horizontal drift from the vertical of lidar station is limited and this supports the assumption that the two sensors are actually sounding the same air masses. The calibration coefficient is obtained through a best-fit procedure applied to the Raman lidar and radiosonde data, the value of the coefficient being the one which minimizes the root mean square deviation between the values of the data points from two profiles within the altitude interval 2-4 km. As the Raman lidar and the radiosonde data have different altitude arrays, for the purpose of applying the best-fit algorithm, radiosonde data have been interpolated to the Raman lidar altitude levels.

This is now much better clarified in the paper, when the following paragraph has been introduced: "The Raman lidar has been calibrated based on an extensive comparison with the radiosondes launched from the nearby station of IMAA-CNR, which is only 8.2 km away from the Raman lidar. Launched radiosondes are manufactured by Vaisala (model: RS92-SGP). For the purpose of determining the calibration constant $c$ the Raman lidar and radiosonde profiles are compared over the altitude interval 2.5-4 km, i.e. above the boundary layer. In fact, in clear sky conditions, the horizontal homogeneity of the humidity field above the boundary layer top is sufficiently high to allow assuming that the Raman lidar and the radiosonde are sounding the same air masses. Within this altitude interval, Raman lidar signals are strong and characterized by high signal-to-noise ratios and small statistical uncertainties. At the same time, within this low level altitude interval, the horizontal drift of the radiosonde with respect to the vertical of lidar station is limited, so that again the two sensors can be actually assumed to be sounding the same air masses. The calibration constant $c$ is obtained through a best-fit procedure applied to the Raman lidar and radiosonde data, the value of the constant being determined by minimizing the root mean square deviation between the single data points from the two profiles within the altitude interval 2.5-4 km. As the Raman lidar and the radiosonde data have different altitude arrays, for the purpose of applying the best-fit algorithm, radiosonde data have been interpolated to the Raman lidar altitude levels.

For the purpose of determining the calibration constant $c$, a specific inter-comparison effort between BASIL and the radiosondes launched from IMAA-CNR was carried out in the period 9 October 2014-7 May 2015. An overall number of 11 comparisons, including all coincident measurements, were possible. In this respect, it is to be specified that routine radiosonde launches started at IMAA-CNR only on October 2014, so inter-comparisons before this date were very infrequent. Figure 2 illustrates the vertical profiles of the water vapour mixing ratio and temperature mean BIAS and RMS deviation for the 11 considered comparisons. The altitude interval 2.5-4 km was used to quantify the mean value of the calibration constant, $\overline{c}$, which is obtained by averaging the single calibration coefficient values from all 11 inter-comparisons. The uncertainty affecting the calibration constant, $\sigma_c$, has been estimated as the standard deviation all single calibration values from the mean value. The value of $\overline{c}$ is found to be equal to 82.33, while the value of $\sigma_c$ is found to be equal to 3.72. The standard deviation, expressed in percentage ($100 \times \sigma_c/\overline{c}$), is found to be

equal to 4.5 %. A very similar procedure was applied to calibrate temperature measurements. In this case the mean value and standard deviation of the calibration constants $a$ and $b$ were determined, with $\overline{a} \pm \sigma_a = 760 \pm 7$ and $\overline{b} \pm \sigma_b = 0.97 \pm 0.03$."

In spite of the interest of such paper, the paper should be improved with respect to the points below:
- Why is the intercomparison limited to the first 2 years of the participation of BASIL to NDACC?

The inter-comparison was limited to the first 2 years of the participation of BASIL to NDACC because, after this period, the laser experienced a long period of reduced emitted power as a results of some unidentified internal optical misalignments, which determined a detriment of the lidar performance. It is to be additionally specified that inter-comparison effort includes only case studies collected in clear sky conditions. In fact, clear sky condition represents the most suited condition for both water vapour and temperature measurements by Raman lidar, with water vapour profile measurements extending up to the UTLS region and temperature profile measurements extending up to 50 km. In this regard it is also to be highlighted that an appropriate assessment of measurement performance based on a sensors/models inter-comparison effort requires the sensors to be operated in clear sky conditions, which is not always the case for either for the Raman lidar or the two passive space sensors IASI and AIRS. More specifically, the Raman lidar system BASIL does not have an all-weather measurement capability, which implies that the system is shut down in case of precipitation. Furthermore, BASIL (and this is true for all lidar systems) cannot penetrate thick clouds, the laser beam being completely extinguished for optical thicknesses around 2. Acceptable Raman lidar performance are still possible above thin clouds, with optical thickness < 0.3. This translates into the fact that, for the purposes of the present inter-comparison effort, even the presence of high cirrus clouds makes case studies non eligible for the comparison. These aspects are now clearly specified in the text, where the following paragraph has been introduced: "For the aims of this paper, we focused our attention on six selected case studies collected during the first 2 years of operation of the system, namely 7 November 2013, 19 December 2013, 9 October 2014, 27 November 2014, and 2 and 9 April 2015. While a larger data-set could have been chosen, we decided to focus our attention only on clear sky cases. In fact, clear sky condition represents the most suited condition for both water vapour and temperature measurements by Raman lidar, with water vapour profile measurements extending up to the UTLS region and temperature profile measurements extending up to 50 km. An appropriate assessment of measurement performance based on a sensors/models inter-comparison effort requires the sensors to be operated in clear sky conditions, which is not always the case for either for the Raman lidar or the two passive space sensors IASI and AIRS. More specifically, the Raman lidar system BASIL does not have an all-weather measurement capability, which implies that the system is shut down in case of precipitation. Additionally, BASIL - and this is true for all lidar systems - cannot penetrate thick clouds, the laser beam being completely extinguished for optical thicknesses around 2. Acceptable Raman lidar performance are still possible above thin clouds, with optical thickness < 0.3. Thus, for the purposes of the present inter-comparison effort, even the presence of high cirrus clouds makes case studies non eligible for the comparison. In other specific case studies IASI and/or AIRS data were characterized by a very poor quality and unrealistic biases, which forced us to remove those from the inter-comparison effort. After April 2015, the laser experienced a period of reduced emitted power, possibly as a results of an unidentified internal optical misalignment. This determined a detriment of the lidar performance, which prevented from considering these measurements within this inter-comparison effort.".

- In the abstract and in the summary, it should be mentioned that the bias values for the entire T and q profiles.

We agree with the reviewer that reporting in the Abstract only vertically-averaged mean biases is misleading. Bias values for the entire temperature and water vapour mixing ratio profiles are now reported in the Abstract, where the following sentences have been introduced: "

Comparisons between BASIL and the different sensor/model data in terms of water vapour mixing ratio indicate mutual biases in the altitude interval 2-15 km always within the interval ± 1 g kg$^{-1}$ (or ± 50 %), with minimum values being observed in the comparison of BASIL vs. radiosondings (± 50 % up to 15 km)." and "For what concerns the comparisons in terms of temperature measurements, results indicate mutual biases in the altitude interval 3-30 km always within the interval ± 3 K, with minimum values being observed in the comparison of BASIL vs. radiosondings (± 2 K within the same altitude interval). Results also reveal mutual biases within ± 3 K up to 50 km for most sensor/model pairs."

Also, how do you reconcile the numbers at lines 20-21, 23 and 26 with those at line 12?

We agree with the reviewer that these numbers may appear conflicting. The different values are now more clearly explained. More specifically, numbers at lines 20-21 of the former version of the paper are now specified to be "mutual biases" between BASIL and all other sensors/models. These are to be distinguished from the numbers at lines 25-26, which represent "overall biases". The estimate of the "overall bias", as clearly specified in section 6.4, is obtained by imposing that the summation of all mutual biases between sensor/model pairs is equal to zero, assuming equal weight for the data reliability of each sensor/model. The choice of attributing equal weight to the data reliability of each sensor/model is driven by the awareness that none of them can *a priori* be assumed more accurate than the others and consequently a "pseudo" reference profile can be obtained by averaging all profiles. The corresponding sentence in the paper has been changed as follows: "The choice of attributing equal weight to the data reliability of each sensor/model is driven by the awareness that none of them can *a priori* be assumed more accurate than the others and thus assuming that the closest profile to a reference profile can be obtained by taking the mean of all profiles."

Furthermore, the numbers at line 12 refer to the "measurement accuracy" and not "measurement bias". This information had been introduced in an incorrect way, without properly justifying and commenting it. Here we refer to "measurement accuracy" and not "measurement bias" not exceeding 0.1 K and 0.1 g kg$^{-1}$", as now properly corrected in the text. Here we use the term "accuracy" to refer to the combined effect on measurement performance of both the random and the systematic error.

- Regarding the vertical resolution of the q profiles: in the abstract and summary it is just mentioned 150 m, whereas in the text in Section 6 (Case study 7 Nov 2013, p14) the resolution is stated to be 300 m between 6 and 8 km and 600 m above 8 km.

The reviewer is right: there was a misprint in the Abstract, which determined an incongruence in the reported values. Now the corresponding sentence in the Abstract reads: "Measurements illustrated in this manuscript demonstrate the ability of BASIL to perform measurements of the temperature profile up to 50 km and of the water vapour mixing ratio profile up to 15 km, when considering an integration time of 2 h and a vertical resolution of 150-**600** m, with measurement mean accuracy, determined based on comparisons with simultaneous and co-located radiosondes, of 0.1 K and 0.1 g kg$^{-1}$, respectively, up to the upper troposphere."

The Same holds for the vertical resolution of the T profiles: in the abstract and summary it is just mentioned 150 m, whereas in the text in Section 6 (p15) the resolution is stated to be 600 m above 6 km.

See comments to the previous point.

- What is the interest of comparing BASIL products with IASI and AIRS products, especially since they are assimilated in NWP model reanalysis products?

We agree that the low values of the mutual bias between the different satellite sensors and the models are partially associated with AIRS, IASI and model data not being independent. This is

specifically true for AIRS and IASI data, these two sensors being operated in a similar way, with their radiance data being analyzed with similar algorithms and average kernels. Additionally, radiance data from these two space sensors are assimilated into the ECMWF and ECMWF-ERA model reanalysis, which make IASI/AIRS data and ECMWF model re-analyses is some extent mutually dependent. **However, this is not true for the mutual biases between the radiosondes and the Raman lidar and between these two sensors and the different satellite sensors and ECMWF model runs**. In fact, radiosondes from IMAA-CNR are not assimilated by ECMWF and the Raman lidar provides completely independent measurements, which are calibrated with unassimilated radiosonde data. These aspects are now better clarified in the paper, where the following sentences have been introduced: "It is to be specified that IASI and AIRS data, together with a variety of additional sensors, are assimilated in ECMWF re-analyses, which makes ECMWF re-analyses partially dependent on IASI and AIRS data, with possible non-negligible effects on the mutual biases between the satellite and the model re-analyses data. However, the mutual biases between radiosondes and the Raman lidar, and between these two sensors and the different satellite sensors and ECMWF re-analyses are completely unaffected by sensor/model cross-dependences, as in fact radiosondes from IMAA-CNR are not assimilated by ECMWF and the Raman lidar provides completely independent measurements, which are calibrated with unassimilated radiosonde data."

- P7: lines 4-5: assessment of K(z) up to 15 km is crucial here to derive the performance of BASIL. You need to say more. How many soundings were used? How do you manage to assess a K(z) up to 15 km with a met sonde that is drifting away from the launch point because of wind?

As already mentioned above, a completely new section (section 6.1) – one page length - has been introduced to illustrate the calibration procedure. The Raman lidar has been calibrated based on an extensive comparison with the radiosondes launched from the nearby station of IMAA-CNR. Launched radiosondes are manufactured by Vaisala (model: RS92-SGP). It is to be specified that this is the most appropriate approach we could consider as in fact the radiosonde launching station is only 8.2 km away from the Raman lidar station and this limited distance gives confidence of the possibility to compare Raman lidar and radiosonde profiles above the boundary layer. In fact, in clear sky conditions, the horizontal homogeneity of the humidity field above the boundary layer top is sufficiently high to allow assuming that the two systems (the Raman lidar and the radiosonde) are sounding the same air masses. More specifically, for the purpose of determining the calibration coefficient the Raman lidar and radiosonde profiles are compared over the altitude interval 2.5-4 km. Within this altitude interval, while we assume water vapour heterogeneity to be small, we can also have strong Raman lidar signals, and consequently high signal-to-noise ratios and small statistical uncertainties. At the same time, within this low level altitude interval, the radiosonde horizontal drift from the vertical of lidar station is limited. This reduces the chances that the two sensors are sounding different air masses. The calibration coefficient (see term "c" in expression below) is obtained through a best-fit procedure applied to the Raman lidar and radiosonde data, the value of the coefficient being determined by minimizing the root mean square deviation between the single data points from the two profiles within the altitude interval 2.5-4 km. As the Raman lidar and the radiosonde data have different altitude arrays, for the purpose of applying the best-fit algorithm, radiosonde data are interpolated to the Raman lidar altitude levels.

The function "K(z)" is the calibration function and not the calibration constant "c", which is determined through a calibration procedure described in detail in Di Girolamo *et al.* (2017) and cited in the manuscript. Specifically, the calibration function "K(z)= c•f(z)" is obtained by multiplying several height-dependent correction terms, included in "f(z)", and the height-independent calibration coefficient "c" (e.g. Whiteman, 2003). The height-dependent correction terms, f(z), are a differential transmission term, accounting for the different atmospheric transmission by molecules and aerosols at the two wavelengths corresponding to the water vapour and molecular nitrogen Raman signals, and a term associated with the use of narrowband interference filters and the consequent temperature dependence of H2O and N2 Raman scattering

signals selected by these filters. The height-independent calibration constant c is finally obtained from the multiplication of the above-mentioned signal ratio (($P_{H2O}(z)$ over $P_{N2}(z)$)) by the height-dependent correction terms, f(z), and the comparison of this quantity with simultaneous and co-located mixing ratio measurements from the radiosondes. As illustrated in a variety of previous papers (among others, Whiteman, 2003), the height-dependent correction terms, f(z) can be determined with an accuracy of 1-3 % from surface up to an altitude of 15 km based on the availability of simultaneous temperature profiles, which are measured by the Raman lidar, and atmospheric number density profiles, which are provided by radiosondes.

The horizontal drift of the radiosonde from the launch point due to the wind does not affect the calibration procedure as in fact the calibration constant c, which obviously has the same value at any altitude, is determined based on the comparison between the Raman lidar and the radiosondes in the altitude interval 2.5-4 km and in this altitude interval the horizontal drift of the radiosonde is assumed to be negligible.

All the above aspects concerning the determination of the calibration function K(z) are now clearly described in the manuscript, where the following sentences have been introduced: "The calibration function *K(z)=c·f(z)* includes an altitude-dependent term *f(z)* associated with the different atmospheric transmission by molecules and aerosols at the two wavelengths corresponding to the water vapour and molecular nitrogen Raman signals and with the use of narrow-band interference filters and the consequent temperature and altitude dependence of $P_{H_2O}(z)$ and $P_{N_2}(z)$ (Whiteman, 2003). *c* is the calibration constant, which is an altitude-independent term obtained from the comparison of the Raman lidar signal ratio $P_{H2O}(z)/P_{N2}(z)$ and, in our specific case, the radiosondes launched from the nearby station of IMAA-CNR. While the calibration procedure applied to BASIL has been illustrated in previous papers (among others, Di Girolamo et al., 2009a,b, 2017), the sensor performance assessment purposes of the present paper impose a proper and detail description of the calibration procedure applied to BASIL before the inter-comparison effort reported in this paper. This is illustrated in section 6.1."

- What kind of humidity sound were used for the RDS? Most (if not all) of the commercial sondes are known to have issues with measurement in low humidity conditions …

The radiosondes considered in the study are manufactured by Vaisala (model: RS92-SGP). These were known as being the most accurate and reliable radiosondes in terms of humidity measurements at the time when the Raman lidar measurements reported in this paper were carried out. In this regard it is also to be specified that the considered radiosondes were launched from the CNR-IMAA Atmospheric Observatory, which is one of the reference stations of GRUAN, the GCOS Reference Upper-Air Network, aimed at providing long-term, highly accurate measurements of the atmospheric profiles, complemented by ground-based state of the art instrumentation, to constrain and calibrate data from more spatially-comprehensive global observing systems. For this purpose, particular care is taken in the selection of the best commercially available sensors, as well as in the operation and processing of the data.

- P8, lines 13-14: a and b are determined from co-located soundings? How do you deal with a met sonde that is drifting away from the launch point because of wind? Up to what altitude 25 km.. how do you ensure a and b are not offset by the soundes drifting? Also what is the sensitivity of the T(z) retrievals on a and b retrievals?

The same arguments used for the calibration of water vapour mixing ratio measurements hold also for the calibration of temperature measurements. Values of the calibration constants a and b are constant with altitude. This values are estimated over the altitude region 2.5-4 km and in this altitude interval the horizontal drift of the radiosonde is assumed to be negligible.

- P9, line 4: The integration technique is designed to retrieve T profiles above 20 km … why do you say below 20 km here?

Obviously, this was a misprint and we thank the reviewer for bringing that to our attention. However, in order to shorten the paper, the paragraph where this sentence was included has now been removed together with this sentence.

- 7 Nov 2013 Case: What do you use to assess the BASIL calibration if there is no RDS?

For this case study, as for all the others, we used the mean value of the calibration constant determined through the procedure illustrated in section 6.1, which refers to 11 comparisons between the Raman lidar and the radiosondes launched from the near-by station of IMAA-CNR. The procedure used to calibrate the data is now clearly and carefully described in the manuscript, where a  completely new section (section 6.1) – one page length - has been introduced.

- Line 21: what kind of smoothing filter?

The vertical smoothing filter we are using here is a simple moving or running average. More specifically, this is a central moving average computed using equally spaced data on either side of the point in the series where the mean is calculated and requires using an odd number of data points in the filter window. This aspect is now clearly specified in the text, where corresponding sentence has been changed as follows: "The considered vertical smoothing filter is a simple central moving or running average computed using equally spaced data (vertical step=30 m) on either side of the point where the mean is calculated, which requires using an odd number of data points in the filter window".

How do you achieve 150 m when the resolution of the 2-h profile is 300 or even 600 m above 6/8 km?

In this specific sentence we refer to the colour map in figure 3b (formerly figure 2b), which shows the time evolution of the water vapour mixing ratio over a 6 h time interval from 16:00 to 22:00 UTC on 7 November 2013. The figure is a succession of 72 consecutive 5-min averaged profiles. For the purpose of reducing signal statistical fluctuations, a vertical smoothing filter was applied to the data achieving an overall vertical resolution of 150 m. The use in the colour map (figure 3b) of a fixed resolution of 150 m throughout the measured interval up to 15 km is motivated by the fact that this kind of illustration of the results (the colour map) is aimed to highlight the water vapour variability up to the middle troposphere, and this is well revealed in the figure. On the contrary, the vertical resolution of 150 m from surface up to 6 km, of 300 m between 6 and 8 km and of 600 m above 8 km is considered for the 2-hour mean water vapour mixing ratio profile measured by BASIL on 7 November 2013 over the time interval 17:00-19:00 UTC, which is illustrated in figure 3a (formerly figure 2a). This figure is aimed to illustrate the performance of the Raman lidar in the UTLS region. For this purpose, it considers a profile averaged over a much longer time interval (2 hours instead of 5 minutes) and a coarser resolution in the upper levels, which allows to increase signal statistics and system performance. In both cases, the different vertical resolutions are obtained by applying a vertical smoothing filter, which is a simple central moving or running average, computed using the equally spaced data (vertical step=30 m) on either side of the point where the mean is calculated. This is now clearly specified in the text.

- P21: Section 6.3, line 23: now the vertical resolution of the profiles is 500 m … not 150 m?

A vertical interval of 500 m is considered for the purpose of computing the biases and RMS deviations for all pairs of inter-compared sensors/models reported in this paper. This is done starting from the equally spaced data points (vertical step=30 m). Thus, 500 m is the amplitude of a vertical window including the data points (17) on which the statistical analysis is applied. This is now more clearly specified in the text, where the following sentence has been introduced: "Considering equally spaced data points with a vertical step of 30 m, the statistical analysis to compute the bias and RMS deviation is applied over 17 data points."

- line 27: what are all sensor/model pairs ? - how many pairs for each type of comparisons? What period does this cover?

The overall number of all possible sensor/model pairs is 15, which is the maximum number of pairs possible when 5 sensors/models are available. More specifically, these are BASIL vs. radisondings (RS), BASIL vs. IASI, BASIL vs. AIRS, BASIL vs. ECMWF, BASIL vs. ECMWF-ERA, RS vs. IASI, RS vs. AIRS, RS vs. ECMWF, RS vs. ECMWF-ERA, IASI vs. AIRS, IASI vs. ECMWF, IASI vs. ECMWF-ERA, AIRS vs. ECMWF, AIRS vs. ECMWF-ERA and ECMWF vs. ECMWF-ERA. For each sensor/model pair we are considering 6 comparisons, one for each of the considered case studies (7 November 2013, 19 December 2013, 9 October 2014, 27 November 2014, and 2 and 9 April 2015). The considered time interval is always the closest in time to the 2 hour integration interval considered for the Raman lidar.

- Section 6.5 p 29: Why only the period 9 October 2014- 7 May 2015? Are 11 comparisons enough? Why not do this for the entire period starting with BASIL entering the NDACC network?

As already specified above, for the purpose of determining the calibration constant $c$, a specific inter-comparison effort between BASIL and the radiosondes launched from IMAA-CNR was carried out in the period 9 October 2014 - 7 May 2015. An overall number of 11 comparisons, including all coincident measurements, were possible. In this respect, it is to be specified that routine radiosonde launches started at IMAA-CNR only on October 2014, so inter-comparisons before this date were very infrequent. This is now clearly specified in the text, where the following sentence has been introduced: "For the purpose of determining the calibration constant $c$, a specific inter-comparison effort between BASIL and the radiosondes launched from IMAA-CNR was carried out in the period 9 October 2014- 7 May 2015. An overall number of 11 comparisons, including all coincident measurements, were possible. In this respect, it is to be specified that routine radiosonde launches started at IMAA-CNR only on October 2014, so inter-comparisons before this date were very infrequent."

[revised manuscript text omitted]